# Elevated pyramidal cell firing orchestrates arteriolar vasoconstriction through COX-2-derived prostaglandin E2 signaling

Benjamin Le Gac[1,2], Marine Tournissac[3], Esther Belzic[1,2], Sandrine Picaud[1,2], Isabelle Dusart[1,2], Hédi Soula[4], Dongdong Li[1,2], Serge Charpak[3], Bruno Cauli[1,2]*

[1]Sorbonne Université, CNRS, Inserm, Neuro-SU, Paris, France; [2]Sorbonne Université, CNRS, Inserm, Institut de Biologie Paris-Seine, Paris, France; [3]Sorbonne Université, CNRS, Inserm, Institut de la Vision, F-75012, Paris, France; [4]Sorbonne Université, INSERM, Nutrition and Obesities: Systemic Approaches, NutriOmics, Research Unit, Paris, France

**\*For correspondence:**
bruno.cauli@upmc.fr

**Competing interest:** The authors declare that no competing interests exist.

## eLife Assessment

This study presents **important** findings on the role of pyramidal cells driving vasoconstriction in brain arteries through a COX-2/PGE2 pathway, with additional contributions from NPY (interneurons) and 20-HETE (astrocytes). Optogenetic stimulation of cortical pyramidal neurons induces vasoconstriction, potentially leading to oxygen and nutrient undersupply in regions with sustained activation - a mechanism potentially relevant under pathological conditions. The authors provide **convincing** evidence from brain slice experiments and some in vivo data from anesthetized animals, carefully discussing the strengths and limitations of both approaches.

**Abstract** Neurovascular coupling, linking neuronal activity to cerebral blood flow, is essential for brain function and underpins functional brain imaging. Whereas mechanisms involved in vasodilation are well-documented, those controlling vasoconstriction remain overlooked. This study unravels the mechanisms by which pyramidal cells elicit arteriole vasoconstriction. Using patch-clamp recording, vascular and $Ca^{2+}$ imaging in mouse cortical slices, we show that strong optogenetic activation of layer II/III pyramidal cells induces vasoconstriction, correlating with firing frequency and somatic $Ca^{2+}$ increase. Ex vivo and in vivo pharmacological investigations indicate that this vasoconstriction predominantly recruits prostaglandin E2 through the cyclooxygenase-2 pathway, and activation of EP1 and EP3 receptors. We also present evidence that specific interneurons releasing neuropeptide Y, and astrocytes, through 20-hydroxyeicosatetraenoic acid, contribute to this process. By revealing the mechanisms by which pyramidal cells lead to vasoconstriction, our findings shed light on the complex regulation of neurovascular coupling.

## Introduction

The brain critically depends on the uninterrupted blood supply provided by a dense vasculature (*Schmid et al., 2019*). Cerebral blood flow (CBF) is locally and temporally controlled by neuronal activity, by an essential process called neurovascular coupling (NVC), and is impaired in early stages of numerous neurological disorders (*Iadecola, 2017*). NVC also serves as the physiological basis for

functional brain imaging widely used to map neuronal activity. Neuronal activity increases CBF within seconds (*Iadecola, 2017*). In the cerebral cortex, the hyperemic response linked to neural activity is supported by dynamically controlled vasodilation that is spatially and temporally constrained by vasoconstriction in a second phase (*Devor et al., 2007*). Conversely, vasoconstriction and decreased CBF usually correlate with reduced neuronal activity (*Devor et al., 2007*; *Shmuel et al., 2002*).

Mounting evidence indicates that the positive correlation between neuronal activity and CBF is not always maintained under physiological conditions: (i) robust sensory-evoked vasodilation can occur in the absence of substantial neuronal response (*O'Herron et al., 2016*), (ii) conversely, pronounced neuronal activity is not systematically associated with increased hemodynamics (*Ma et al., 2016*), (iii) CBF is decreased in several cortical areas despite local increase in neuronal activity (*Devor et al., 2008*), and (iv) optogenetic simulation of inhibitory GABAergic interneurons results in vasodilation (*Uhlirova et al., 2016*). Furthermore, in pathological conditions with intense neuronal activity such as epileptic seizures, a sustained hypoperfusion induced by vasoconstriction is observed (*Farrell et al., 2016*; *Tran et al., 2020*).

NVC is achieved by the synthesis and release of vasoactive messengers within the neurovascular unit (*Iadecola, 2017*), which act on the contractility of mural cells (smooth muscle cells and pericytes) to control vessel caliber and CBF along the vascular tree (*Rungta et al., 2018*). Pial and penetrating arterioles, which have a higher density of contractile mural cells and control their diameter faster than capillaries (*Hartmann et al., 2021*; *Hill et al., 2015*; *Rungta et al., 2021*; *Rungta et al., 2018*), play a key role in regulating CBF.

Different experimental approaches, each with their advantages and limitations, have allowed the identification of several mediators of NVC (*Grutzendler and Nedergaard, 2019*; *Iadecola and Nedergaard, 2007*). Ex vivo brain slices provide a well-controlled environment, ideal for pharmacological investigations to dissect the underlying mechanisms. However, they lack connectivity and blood flow which provides both vascular tone and natural oxygenation and therefore require in vivo validation. Conversely, pharmacological studies are more challenging with in vivo preparations. Awake animals allow physiologically relevant context with largely undisturbed network and neuromodulatory activity. However, this preparation is subject to brain state changes which may affect network activity, metabolism, and vascular physiology (*Grutzendler and Nedergaard, 2019*), potentially complexifying the analysis of specific mechanisms of NVC. Although chronic anesthetized animals have reduced network and neuromodulation activity, the NVC response is only slowed (*Rungta et al., 2021*), providing a valuable model for validating ex vivo observations.

Messengers of vasodilation released by excitatory neurons, GABAergic interneurons, astrocytes, or endothelial cells, include nitric oxide, $K^+$, arachidonic acid derivatives such as prostaglandin E2 (PGE2; *Iadecola, 2017*), or more recently glutamate (*Zhang et al., 2024*). Despite its physio pathological importance, vasoconstriction is less understood with fewer cell types and vasoactive messengers that have been identified. It is now generally accepted that GABAergic interneurons are key players in vasoconstriction by releasing neuropeptide Y (NPY; *Cauli et al., 2004*; *Uhlirova et al., 2016*). Under certain conditions, astrocytes can also induce vasoconstriction via 20-hydroxyeicosatetraenoic acid (20-HETE) (*Mulligan and MacVicar, 2004*) or high $K^+$ concentration (*Girouard et al., 2010*). However, the involvement of pyramidal cells in vasoconstriction has been overlooked.

PGE2 has emerged as a bimodal messenger of NVC, similar to $K^+$ (*Girouard et al., 2010*) and glutamate (*Zhang et al., 2024*), that can induce either vasodilation (*Gordon et al., 2008*; *Lacroix et al., 2015*; *Lecrux et al., 2011*; *Mishra et al., 2016*) or vasoconstriction (*Dabertrand et al., 2013*; *Rosehart et al., 2021*) depending on its concentration and/or site of action along the vascular tree. Under physiological conditions, PGE2 is produced during NVC by either astrocytes (*Mishra et al., 2016*) or pyramidal cells (*Lacroix et al., 2015*) via the rate-limiting synthesizing enzymes cyclooxygenase-1 (COX-1) or –2 (COX-2), respectively. Since COX-2-expressing pyramidal cells can release glutamate and PGE2, both of which induce vasoconstriction at high concentrations (*Dabertrand et al., 2013*; *Rosehart et al., 2021*; *Zhang et al., 2024*), pyramidal cells may be responsible for vasoconstriction when their spiking activity is high.

To test this hypothesis, we used ex vivo and in vivo approaches in combination with optogenetics to precisely control pyramidal cell firing in the mouse barrel cortex while monitoring the resulting arteriolar response. We found that pyramidal cells induce vasoconstriction at high stimulation frequency and about half of them express all the transcripts required for a cell autonomous synthesis of the

vasoconstrictor messengers PGE2 and prostaglandin F2α (PGF2α). Pharmacological investigations revealed that this neurogenic vasoconstriction depends on COX-2-derived PGE2 via the direct activation of vascular EP1 and EP3 receptors. It also involves the recruitment of intermediary NPY interneurons acting on the Y1 receptor, and, to a lesser extent astrocytes, via 20-HETE and COX-1-derived PGE2. Thus, our study reveals the mechanisms by which high-frequency pyramidal cell firing leads to vasoconstriction.

## Results

### Pyramidal cells induce vasoconstriction at high firing frequency

To determine if pyramidal cells action potential (AP) firing can induce vasoconstriction in a frequency-dependent manner, we used optogenetics to induce AP firing while monitoring the resulting vascular response in cortical slices. We used Emx1-cre;Ai32 transgenic mice expressing the H134R variant of channelrhodopsin-2 (ChR2) in the cortical glutamatergic neurons (*Gorski et al., 2002*), conferring robust pyramidal cell photoexcitability (*Madisen et al., 2012*). Wide-field photostimulation of cortical slices was achieved in layers I to III (*Figure 1—figure supplement 1A*) using 10 s trains of 5ms light pulses (see Materials and methods) delivered at five different frequencies (1, 2, 5, 10 and 20 Hz, *Figure 1A*).

First, we ensured the efficiency of the photostimulation paradigm by recording layer II-III pyramidal cells in whole-cell current clamp mode (*Figure 1A*). We observed that optogenetic stimulation resulted in the firing of an initial AP that was followed by a train of spikes whose amplitude and frequency transiently decreased before reaching a steady state (*Figure 1A*, upper traces). Consistent with the kinetic properties of the H134R ChR2 variant (*Lin et al., 2009*) and the intrinsic firing properties of pyramidal cells (*Karagiannis et al., 2009*), the steady-state firing frequency matched the photostimulation frequency up to 5 Hz but was lower at higher frequencies (*Figure 1A*, steady-state spike success rate: 100 ± 0% at 1, 2, and 5 Hz, 70 ± 11% at 10 Hz and 55 ± 12% at 20 Hz). These observations demonstrate efficient pyramidal cell activation over a wide range of photostimulation frequencies.

To test the hypothesis that neuronal activity induces vasoconstriction, we analyzed the optogenetically induced response of penetrating arterioles. Layer I arterioles were imaged for 30 min in cortical slices (*Figure 1—figure supplement 2*; *Table 1*) without preconstriction to facilitate observation of vasoconstriction (*Cauli et al., 2004*). Examination of the evoked vascular response over 30 min (*Figure 1—figure supplement 2*) showed that increasing the frequency of photostimulation shifted the overall vascular response from a barely discernible delayed response between 1 Hz and 5 Hz to a sustained vasoconstriction at 10 Hz and above which began less than 2 min after photostimulation (10 Hz: 1.4±0.4 min; 20 Hz: 1.6±0.5 min). Most vessels (n=8 of 10 arterioles) showed a strong and rapid vasoconstriction at 20 Hz. On average, this response peaked at 6.8±2.4 min, much earlier than at lower frequencies, which typically required more than 10 min to reach a maximum (*Figure 1—figure supplement 2C*, 1 Hz: 15.6±4.0 min; 2 Hz: 13.2±2.3 min; 5 Hz: 16.0±3.6 min; 10 Hz: 15.7±2.4 min). Because the vascular response shifted to reliable vasoconstriction, with onset and peak in less than 2 and 10 min, respectively, similar to previous observations in cortical slices (*Cauli et al., 2004*), when the frequency of photostimulation was increased to 20 Hz, we defined the first 10 min of recording as the vasoconstriction time frame for subsequent comparisons and analyses. While photostimulation at 1–5 Hz failed to elicit fast reliable vascular responses (*Figure 1C and D*), 10 Hz photostimulation predominantly induced vasoconstriction (n=4 of 5 arterioles, *Figure 1C and D*, area under the curve (AUC)=$-1.7 \pm 1.1 \times 10^3$ %.s, n=5). This response was even more pronounced at 20 Hz, as all arterioles showed vasoconstriction of high magnitude (*Figure 1B–D*; AUC = $-3.7 \pm 0.7 \times 10^3$ %.s, $F_{(4, 30)}$=6.135, p=9.89 x $10^{-4}$, one-way ANOVA, n=10 arterioles). This difference was particularly striking when comparing the magnitude at 20 Hz (*Figure 1D*) with those at 1 Hz ($t_{(12)}$ = $-3.48$, p=0.0407, t-test), 2 Hz ($t_{(18)}$ = $-4.09$, p=0.0250, t-test) and 5 Hz ($t_{(14)}$ = $-3.7$, p=0.0346, t-test). Intense optogenetic stimulation of pyramidal cells has been shown to elicit cortical spreading depression (*Chung et al., 2019*; *Pham et al., 2024*), which induces vasoconstriction (*Zhang et al., 2024*) and fast cell swelling (*Zhou et al., 2010*). We ruled out this possibility by showing that the rate of change in light transmittance associated with cell swelling remained below that of cortical spreading depression (*Zhou et al., 2010*; *Table 1*). On the other hand, ChR2-independent vascular changes induced by high light

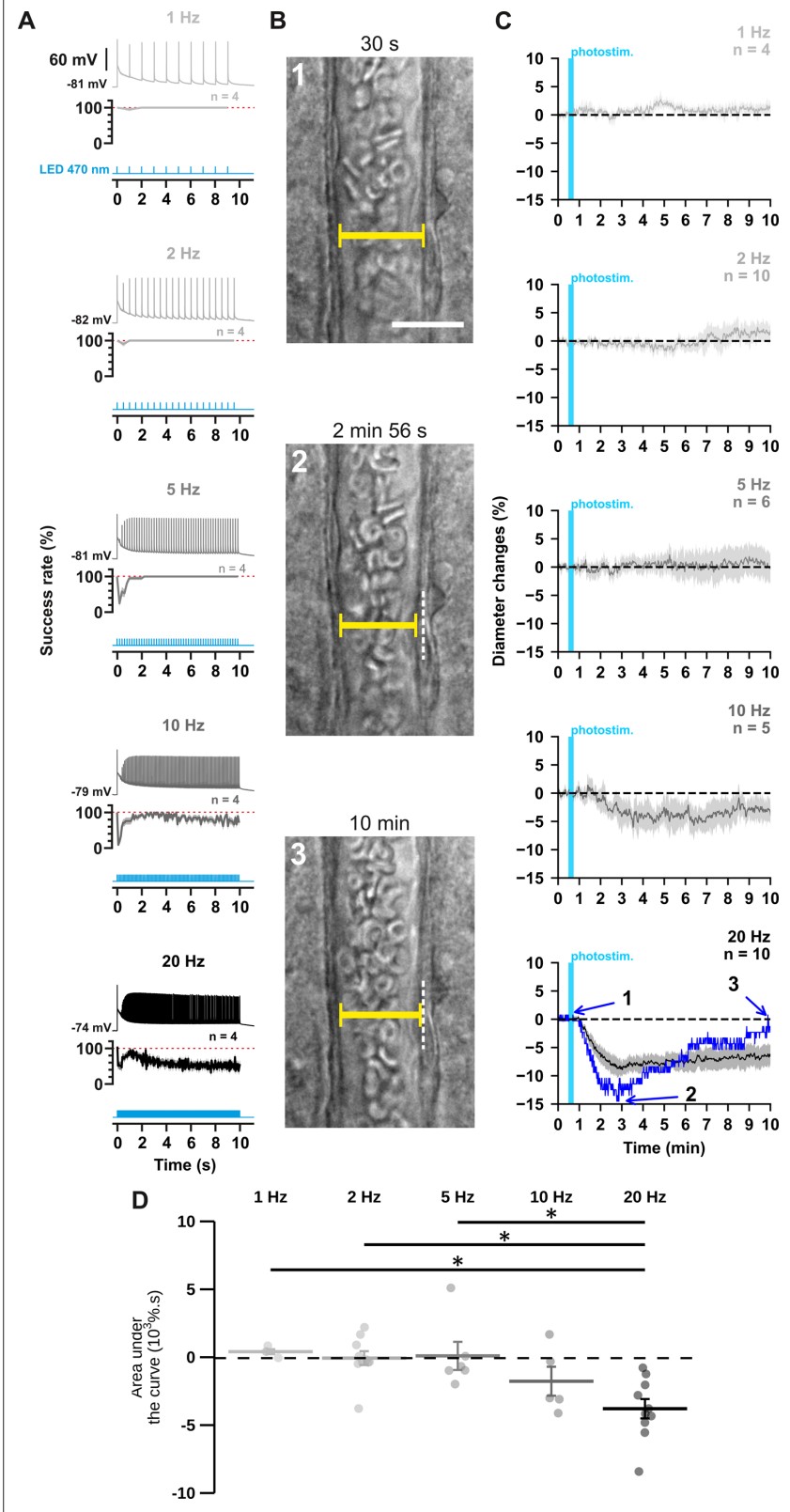

**Figure 1.** The occurrence and strength of vasoconstriction depends on the photostimulation frequency of pyramidal cells. (**A**) Representative examples of the voltage responses of a layer II-III pyramidal cell (upper traces light grey to black traces) induced by photostimulations (470 nm, 10 s train, 5ms pulses) delivered at 1, 2, 5, 10, and 20 Hz (cyan lower traces) and mean spike success rate (middle trace, n=4 cells from 3 mice). The SEMs envelope

*Figure 1 continued*

the mean traces. The red dashed lines represent a spike success rate of 100%. (**B**) Representative example showing IR-DGC pictures of a layer I penetrating arteriole (1) before a 20 Hz photostimulation, (2) at the maximal diameter decrease, and (3) after 10 min of recording. Pial surface is upward. Yellow calipers represent the measured diameters. White dashed lines indicate the initial position of the vessel wall. Scale bar: 25 µm. (**C**) Kinetics of arteriolar diameter changes induced by photostimulation (vertical cyan bars) at 1 Hz (n=4 arterioles from 3 mice), 2 Hz (n=10 arterioles from 8 mice), 5 Hz (n=6 arterioles from 6 mice), 10 Hz (n=5 arterioles from 5 mice), and 20 Hz (n=10 arteriole from 9 mice). The SEMs envelope the mean traces. The blue trace represents the kinetics of the diameter changes of the arteriole shown in (**B**). (**D**) Effects of the different photostimulation frequencies on AUC of vascular responses during 10 min of recording. Data are presented as the individual values and mean ± SEM. * statistically different from 20 Hz stimulation with p<0.05.

The online version of this article includes the following source data and figure supplement(s) for figure 1:

**Source data 1.** Detection of spikes per light pulse interval from different cells used to determine spike success rate in *Figure 1A*.

**Source data 2.** Diameter measurements (µm) of individual arterioles used to determine diameter changes in *Figure 1C* and *Figure 1—figure supplement 2*.

**Figure supplement 1.** Vasoconstriction induced by widefield photostimulation is specific of ChR2 expression in pyramidal cells.

**Figure supplement 1—source data 1.** Diameter measurements (µm) of individual arterioles used to determine diameter changes in *Figure 1—figure supplement 1C*.

**Figure supplement 2.** Vasoconstrictions occurred during the 30 first minutes after pyramidal cells photoactivation.

intensity have been reported (*Rungta et al., 2017*). We verified that 20 Hz photostimulation did not induce a vascular response in wild-type mice that do not express ChR2 (*Figure 1—figure supplement 1*). Taken together, our observations indicate that photostimulation of pyramidal cells produces a frequency-dependent vasoconstriction.

**Table 1.** Morphological and physiological properties, and neurovascular responses of diving arterioles used in the analysis of the frequency-dependence of the polarity of neurovascular response evoked by pyramidal cells.

| Frequency | 1 Hz | 2 Hz | 5 Hz | 10 Hz | 20 Hz |
|---|---|---|---|---|---|
| Number of arterioles | n=4 | n=10 | n=6 | n=5 | n=10 |
| | 1.2±0.2 | 1.6±0.2 | 1.6±0.2 | 1.3±0.2 | 1.0±0.1 |
| | $F_{(4, 30)}=2.161$ $p=0.098$ | | | | |
| Resting stability (%) | n.s. | | | | |
| | 3.6±0.8 | 3.8±0.3 | 3.2±0.5 | 4.1±0.8 | 4.0±0.2 |
| | $F_{(4, 30)}=0.656$ $p=0.627$ | | | | |
| Wall thickness (µm) | n.s. | | | | |
| | 0.5±0.2 | 0.0±0.5 | 0.2±1 | −1.7±1.1 | −3.7±0.7 |
| | $F_{(4, 30)}=6.135$ $p=0.00099$ | | | | |
| Area under the curve after photostimulation (AUC; $\times 10^3$ %.s) | *** | | | | |
| | 0.19±0.09 | 0.64±0.14 | 0.22±0.04 | 0.43±0.19 | 0.8±0.11 |
| Maximal $d\Delta T/dt$ (%.s$^{-1}$) | All <2 %.s$^{-1}$ | | | | |

Data are mean ± SEM, one-way ANOVA F test and corresponding exact p-value. n.s., not statistically different and ***: p<0.001.

The online version of this article includes the following source data for table 1:

**Source data 1.** Properties of individual arterioles used for *Table 1*.

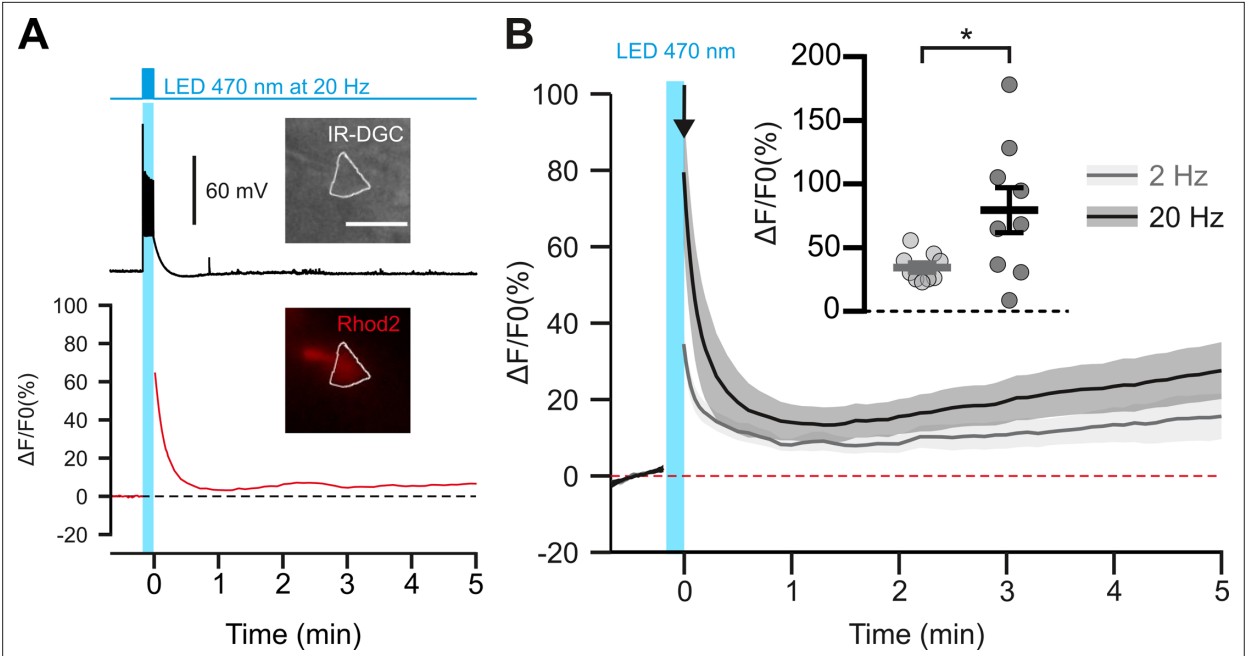

**Figure 2.** Photostimulation of pyramidal cells elicits a time-locked firing and a frequency-dependent calcium increase. (**A**) Voltage response (top trace) and kinetics of relative fluorescence changes (red bottom trace) induced by photostimulation at 20 Hz. Insets, IR-DGC (top), Rhod2 fluorescence (bottom) pictures of an imaged layer II/III pyramidal cell. The somatic region of interest is outlined in white. Pial surface is upward. Scale bar: 20 μm. (**B**) Mean relative variations of $Ca^{2+}$ fluorescence in response to photostimulation at 2 Hz (grey, n=9 cells from 5 mice) and 20 Hz (black, n=9 cells from 5 mice). Dashed line represents the baseline. The vertical cyan bar indicates the duration of photostimulation. SEMs envelope the mean traces. Inset, Maximum increase in relative fluorescence changes induced immediately after photostimulation, indicated by the black arrow. The data are shown as the individual values and mean ± SEM. * statistically different with p<0.05.

The online version of this article includes the following source data and figure supplement(s) for figure 2:

**Source data 1.** Somatic fluorescence measurements (A.U.) used to determine fluorescence changes in *Figure 2*.

**Figure supplement 1.** Photostimulation of pyramidal cells does not evoke recurrent spiking network activity.

**Figure supplement 1—source data 1.** Detection of spikes per second used to determine the mean firing frequency in *Figure 2—figure supplement 1C and D*.

## Optogenetic stimulation induces a frequency-dependent, gradual increase in somatic calcium that precedes the vascular response

These observations raise the questions of how pyramidal neurons can induce vasoconstriction at higher AP-firing rates. It is generally accepted that the synthesis and/or release of vasodilatory substances requires an increase in intracellular $Ca^{2+}$ in the releasing cells (*Attwell et al., 2010*; *Cauli and Hamel, 2010*), but little is known about the release of vasoconstricting substances. We therefore determined whether an increase in somatic $Ca^{2+}$ concentration in cortical neurons was also dependent on photo-stimulation frequency. We combined optogenetic stimulation with whole-cell current clamp recording and intracellular $Ca^{2+}$ imaging using Rhod-2 delivered by patch pipette (*Figure 2A*). Excitation of this red $Ca^{2+}$ indicator at 585 nm did not induce any voltage response in the recorded pyramidal cells (*Figure 2A*), as expected from the action spectrum of ChR2 (*Lin et al., 2009*). In contrast, photostim-ulation at 470 nm elicited a train of spikes accompanied by a somatic $Ca^{2+}$ increase that decayed for tens of seconds after photostimulation without triggering any significant recurrent spiking activity (*Figure 2—figure supplement 1*). The $Ca^{2+}$ response evoked by 20 Hz photostimulation was more than twice of that evoked by 2 Hz photostimulation (*Figure 2B*; 2 Hz: $\Delta F/F_0$=34.5 ± 3.7 %, n=9 cells, vs. 20 Hz:$\Delta F/F_0$=79.5 ± 17.7 %, n=9 cells; $t_{(16)}$=2.485, p=0.024397), while the average number of evoked spikes was about five times higher (*Figure 1A*, *Figure 2—figure supplement 1*). These results demonstrate a frequency-dependent increase in intracellular $Ca^{2+}$ induced by photostimulation, that precedes vasoconstriction. We therefore aimed to understand the molecular mechanisms linking neuronal activity to vasoconstriction.

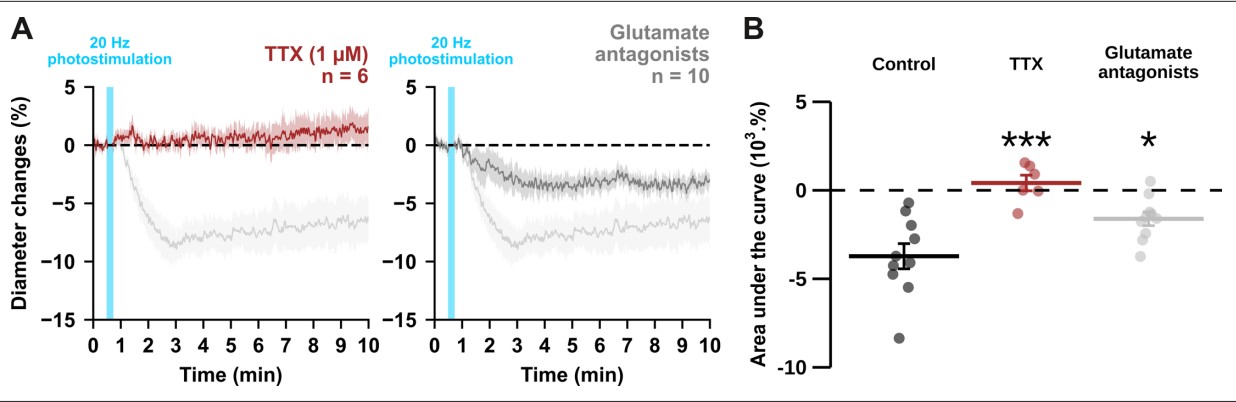

**Figure 3.** Optogenetically-induced vasoconstriction requires AP firing and partially glutamatergic transmission. Effect of TTX (1 μM, brown, n=6 arterioles from 5 mice) and cocktail antagonists of AMPA/kainate (DNQX, 10 μM), NMDA (D-AP5, 50 μM), mGluR1 (LY367385, 100 μM) and mGluR5 (MPEP, 50 μM) receptors (gray, n=10 arterioles from 6 mice) on (**A**) kinetics and (**B**) magnitude of arteriolar vasoconstriction induced by 20 Hz photostimulation (cyan bar). The SEMs envelope the mean traces. Dashed lines represent the initial diameter. The shaded traces correspond to the kinetics of arteriolar vasoconstriction in control condition (**Figure 1C** – 20 Hz). Data are presented as the individual values and mean ± SEM. * and *** statistically different from control condition (**Figure 1D** – 20 Hz) with p<0.05 and p<0.001, respectively.

The online version of this article includes the following source data and figure supplement(s) for figure 3:

**Source data 1.** Diameter measurements (μm) of individual arterioles used to determine diameter changes in **Figure 3A**.

**Figure supplement 1.** Basal network activity and tonic glutamate do not influence resting vascular tone.

**Figure supplement 1—source data 1.** Diameter measurements (μm) of individual arterioles used to determine diameter changes in **Figure 3—figure supplement 1**.

## Vasoconstriction induced by pyramidal cells requires AP firing and is partially dependent on glutamatergic transmission

In pyramidal cells, APs induce both somatic $Ca^{2+}$ elevation (**Smetters et al., 1999**) and glutamate release. To determine whether spiking activity is required for vasoconstriction induced by 20 Hz photostimulation, we blocked APs with the voltage-activated sodium channel blocker tetrodotoxin (TTX, 1 μM, n=6 arterioles). This treatment completely abolished the vasoconstriction evoked by 20 Hz photostimulation (**Figure 3**; AUC = 0.4 ± 0.4 x $10^3$ %.s, $t_{(14)}$ = 5.57, p=8.6656 x $10^{-6}$). These data indicate that APs are mandatory for neurogenic vascular response and may involve glutamate release.

Indeed, high levels of glutamate released from pyramidal cells may activate NPY-expressing interneurons or astrocytes through activation of ionotropic or group I metabotropic glutamate receptors (**Girouard et al., 2010**; **Mulligan and MacVicar, 2004**; **Uhlirova et al., 2016**). It may also directly activate NMDA receptors on arteriolar smooth muscle cells, resulting in a large intracellular $Ca^{2+}$ increase and subsequent vasoconstriction (**Zhang et al., 2024**). To test the hypothesis that glutamate from pyramidal cells, either directly or indirectly, results in vasoconstriction, we blocked glutamatergic transmission by antagonizing AMPA/kainate, NMDA and group I metabotropic receptors expressed by cortical neurons (**Tasic et al., 2016**; **Zeisel et al., 2015**) and juvenile astrocytes (**Sun et al., 2013**). Glutamate receptor antagonists reduced the magnitude of vasoconstriction (–1.6±0.4 x $10^3$ %.s, $t_{(18)}$ = 3.28, p=0.0160) by approximately half (**Figure 3**). Taken together, our data suggest that photostimulation of pyramidal cells elicits a frequency-dependent vasoconstriction that requires AP firing and partially involves glutamatergic transmission. We therefore sought to elucidate the glutamate-independent vasoactive pathway underlying this neurogenic vascular response.

## Pyramidal cells express the mRNAs for a cell autonomous PGE2 and PGF2α synthesis

Several arachidonic acid metabolites produced after intracellular $Ca^{2+}$ elevation, including PGF2α, but also PGE2, exert dose-dependent vasoconstrictive effects (**Dabertrand et al., 2013**; **Rosehart et al., 2021**; **Zonta et al., 2003**). These prostaglandins could therefore be progressively released as the frequency of photostimulation and somatic $Ca^{2+}$ increase and thereby promote vasoconstriction. Layer II-III pyramidal cells have been shown to produce PGE2 (**Lacroix et al., 2015**). To determine whether

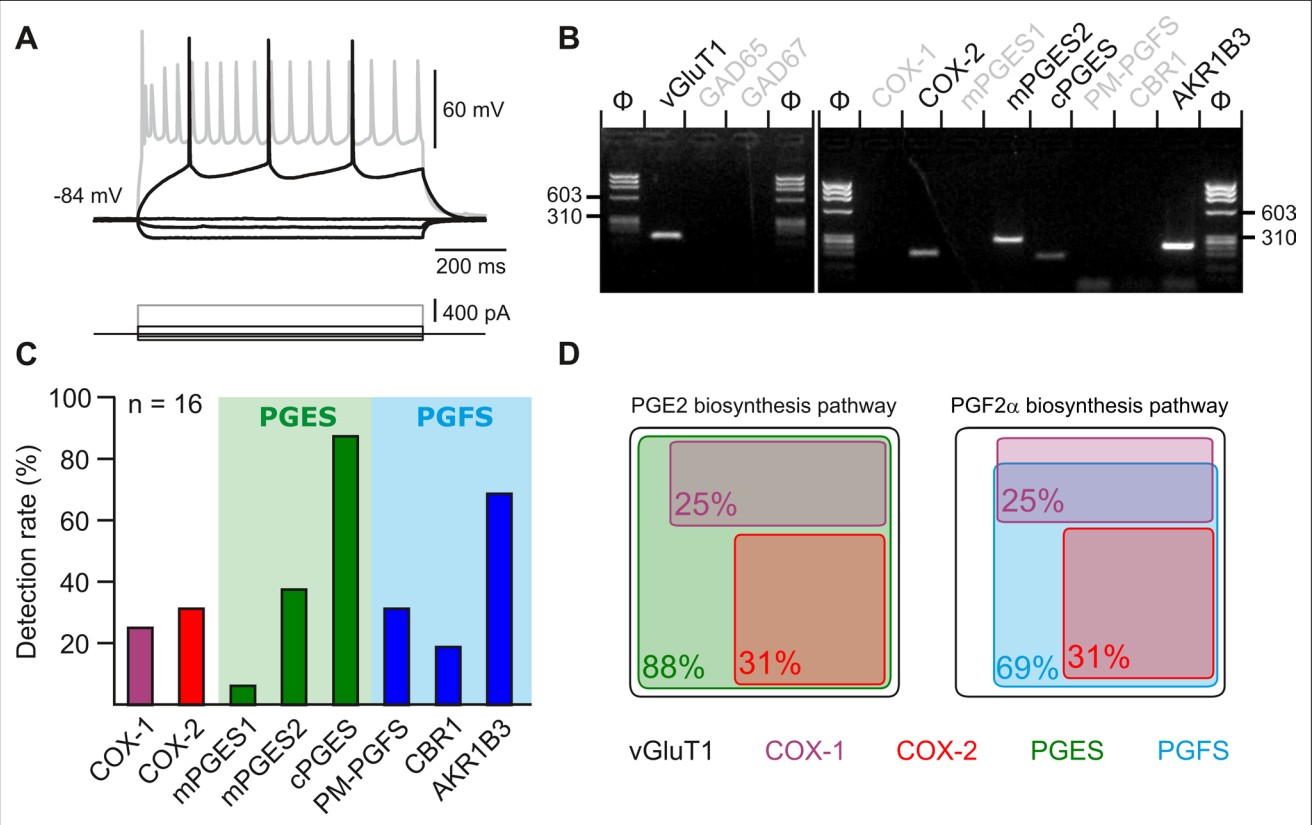

**Figure 4.** Layer II-III pyramidal cells express PGE2 and PGF2α synthesizing enzymes. (**A**) Voltage responses of a layer II-III pyramidal cell induced by injection of current (bottom traces). In response to a just-above-threshold current pulse, the neuron fired long-lasting action potentials with little frequency adaptation (middle black trace). Near saturation, it exhibits the pronounced spike amplitude accommodation and marked frequency adaptation characteristic of regular spiking cells (upper grey trace). (**B**) Agarose gel analysis of the scRT-PCR products of the pyramidal cell shown in (**A**) revealing expression of vGluT1, COX-2, mPGES2, cPGES, PM-PGFS and CBR1. Φx174 digested by *HaeIII* (Φ in bps) was used as molecular weight marker (**C**) Histogram summarizing the single-cell detection rate of PGE2 and PGf2α synthesizing enzymes in layer II-III pyramidal cells (n=16 cells from 6 mice). PGES (green zone) corresponds to mPGES1, mPGES2 and/or cPGES and PGFS (blue zone) to PM-PGFS, CBR1 and/or AKR1B3. (**D**) Co-expression of PGE2 and PGf2α synthesizing enzymes in pyramidal cells. The box size is proportional to the detection rate. Note the absence of co-expression between COX-1 (purple) and COX-2 (red). Co-expression of a PGES (left, green) and a PGFS (right, blue) with COX-1 (up) and COX-2 (bottom).

The online version of this article includes the following source data and figure supplement(s) for figure 4:

**Source data 1.** Original files of the full raw unedited gels shown in *Figure 4B*.

**Source data 2.** Uncropped gels shown in *Figure 4B* with relevant lanes labeled.

**Figure supplement 1.** Sensitivity of the RT-mPCR protocol.

**Figure supplement 1—source data 1.** Original file of the full raw unedited gel shown in *Figure 4—figure supplement 1*.

**Figure supplement 1—source data 2.** Uncropped gels shown in *Figure 4—figure supplement 1* with relevant lanes labeled.

the synthesizing enzymes of PGE2 and PGF2αare present in pyramidal cells, we performed single-cell RT-PCR after patch-clamp recording (*Devienne et al., 2018*). Sixteen layer II-III pyramidal cells were visually identified based on the triangular shape of their soma and a prominent apical dendrite. Their glutamatergic phenotype was confirmed both by their stereotypical regular spiking firing pattern (*Figure 4A*) and also by the expression of the vesicular glutamate transporter, vGlut1, and neither of the two GABA synthesizing enzymes, thus excluding possible contamination by GABAergic interneurons (*Figure 4B*; *Karagiannis et al., 2009*). The rate-limiting enzymes of prostaglandin synthesis, cyclooxygenase-1 (COX-1) and –2 (COX-2), were detected in 25% (n=4 of 16 cells) and 31% (n=5 of 16 cells) of pyramidal cells, respectively, (*Figure 4B–D*) but were never co-expressed (*Figure 4D*). Although the differential expression of COX-1 or COX-2 allowed the definition of three non-overlapping molecular subpopulations of pyramidal cells, they did not show distinctive electrophysiological features

(*Table 2*). The cytosolic enzyme responsible for synthesizing PGE2 (cPGES) was observed in most pyramidal cells (*Figure 4B and C*; 88%, n=14 of 16 cells). In addition, the microsomal PGES, mPGES1 and mPGES2 were detected in 6% (*Figure 4C*, n=1 of 16 cells) and 38% (*Figure 4B and C*; n=6 of 16 cells) of pyramidal cells, respectively, and were always co-expressed with cPGES. The PGF2α terminal-synthesizing enzyme AKR1B3 was observed in the majority of neurons (*Figure 4B and C*; n=11 of 16 cells, 69%). Occasionally, it was co-detected with the prostamide/prostaglandin F synthase (PM-PGFS, *Figure 4C*, n=5 of 16 cells, 31%) and the PGE2 converting enzyme carbonyl reductase 1 (CBR1, *Figure 4C*; n=3 of 16 cells, 19%). CBR1 was consistently detected alongside at least one PGES. Pyramidal cells positive for COX-1 also expressed PGES (*Figure 4D*), with most of them also co-expressing PGFS (n=3 out of 4). All neurons positive for COX-2 co-expressed both PGES and PGFS (*Figure 4D*). These molecular observations suggest that subpopulations accounting for about half of layer II-III pyramidal cells express all the transcripts necessary for the synthesis of PGE2 and PGF2α.

## Prostaglandins underpin vasoconstriction ex vivo and in vivo

To investigate whether prostaglandins could mediate neurogenic vasoconstriction, we inhibited their synthesis. In cortical slices, the non-selective COX inhibitor indomethacin (5 μM, *Table 3*, *Figure 5A and B*) completely abolished the vascular response (n=10 arterioles, AUC = 0.0 ± 0.2 x $10^3$ %.s, $t_{(18)}$ = 5.86, *P*=3.4385 x $10^{-5}$). To verify that high-frequency stimulation of pyramidal cells also induces vasoconstriction in vivo, 10 Hz photostimulation was reproduced in anesthetized Emx1-cre;Ai32 mice. Pial arterioles diameter measured by two-photon line-scan imaging (*Figure 5C and D*) revealed that pyramidal cells induce vasodilation (1st phase) followed by sustained vasoconstriction (2nd phase, *Figure 5D and E*). The constriction phase was inhibited by indomethacin (injected intravenously (i.v.)), indicating the involvement of prostaglandins in the vasoconstriction ($AUC_{Ctrl}$ = .291.5 ± 92.4 %.s vs. $AUC_{Indo.}$=332.4 ± 184.4 %.s, $U_{(5,4)}$ = 0, p=0.0159, *Figure 5D–F*), which confirms our ex vivo observations. To determine whether they originated from COX-1 or COX-2 activity, we utilized selective inhibitors in cortical slices. The vasoconstriction magnitude was reduced by the COX-1 inhibitor SC-560 (100 nM, *Table 3*, n=10 arterioles, –1.4±0.7 x $10^3$ %.s, $t_{(18)}$ = 3.54, p=9.3396 x $10^{-3}$, *Figure 5A and B*). The COX-2 inhibitor NS-398 (10 μM, *Table 3*, n=7 arterioles) completely abolished pyramidal cell-induced vasoconstriction in a more potent manner (*Figure 5A and B*; AUC = 0.1 ± 0.3 x $10^3$ %.s, $t_{(15)}$ = 5.45, p=1.0853 x $10^{-5}$), mimicking the ex vivo effect of indomethacin. These observations suggest that prostaglandins, derived mainly from COX-2 activity, and to a lesser extent from COX-1 activity, mediate pyramidal cell-induced vasoconstriction.

## PGE2 mediates vasoconstriction by acting primarily on EP1 receptor

To determine the nature of the prostaglandins and their receptors, we selectively antagonized the vasoconstrictor receptors of PGE2, EP1 or EP3, or the FP receptor of PGF2α. The magnitude of vasoconstriction was reduced by the selective EP1 receptor antagonist ONO-8130 (10 nM, *Table 3*, n=9 arterioles, *Figure 5G and H*, 0.3±0.3 x $10^3$ %.s, $t_{(17)}$ = 6.01, p=2.8451 x $10^{-6}$), and to a lesser extent, by the EP3 receptor antagonist L-798,106 (1 μM, *Table 3*, n=9 arterioles, *Figure 5G and H*, –0.9±0.4 x $10^3$ %.s, $t_{(17)}$ = 4.30, p=8.0261 x $10^{-4}$). Impairing FP receptor signaling with AL-8810 (10 μM, *Table 3*, n=9 arterioles, *Figure 5G and H*) tended to reduce the evoked vasoconstriction, however, it did not reach statistical significance (AUC = –1.9 ± 0.4 x $10^3$ %.s, $t_{(17)}$ = 2.82, p=0.0533). Additionally, the preconstricted state induced by this weak partial FP agonist (*Sharif and Klimko, 2019*; *Figure 5—figure supplement 1F*) resulted in a diameter reduction of approximately 4% (diameter before application 21.8±2.5 μm vs. during 20.9±2.6 μm, n=9 arterioles, $t_{(8)}$ = 2.374, p=0.0457, paired t-test), which underestimated the optogenetic vascular response. Taken together, these results indicate that pyramidal cell photoactivation induces vasoconstriction through the release of PGE2 originating mainly from COX-2. This effect primarily acts on the EP1 receptor and, to a lesser extent, on the EP3 receptor.

To test a direct effect of PGE2 through vascular EP1/EP3 activation, we determined whether exogenous agonists of PGE2 receptors could mimic the vasoconstriction induced by pyramidal cell photostimulation. Similar to increasing photostimulation frequencies, exogenous application of PGE2 induced vasoconstriction in a dose-dependent manner which persisted for several minutes after removal (*Figure 5—figure supplement 2A and D*). Likewise, 10 μM sulprostone, an EP1/EP3 agonist with an $EC_{50}$ comparable to that of PGE2 (*Boie et al., 1997*), mimicked the vasoconstriction induced by 1–10 μM PGE2 (*Figure 5—figure supplement 2*). Application of 10 μM PGE2 in the presence of

**Table 2.** Electrophysiological properties of pyramidal cells recorded during single-cell RT-PCR experiments.

| | COXs-negative (n=7) | COX-1 positive (n=4) | COX-2 positive (n=5) |
|---|---|---|---|
| **Passive properties** | | | |
| Resting potential (mV) | –82.0±2.2 | –84.7±4.2 | –82.0±6.1 |
| Input resistance (MΩ) | 329±53.7 | 360.8±63.7 | 314.2±79.7 |
| Time constant (ms) | 50.7±6.4 | 47.3±7.2 | 47.74±10.1 |
| Membrane capacitance (pF) | 161.4±14.7 | 133.2±7.8 | 159.4±23.0 |
| Sag index (%) | 11.3±3.8 | 6.7±1.1 | 6.9±1.8 |
| **Just above threshold properties** | | | |
| Rheobase (pA) | 52.7±8.7 | 51.7±15.5 | 62.3±18.2 |
| First spike latency (ms) | 295.2±50.8 | 271.6±62.9 | 178.7±55.6 |
| Adaptation (Hz/s) | –3.1±0.9 | –2.6±0.3 | –3.3±1.3 |
| Minimal frequency (Hz) | 5.5±0.7 | 4.6±0.4 | 6.3±1.2 |
| **Firing properties** | | | |
| Accommodation (mV) | 16.4±4.6 | 24.8±4.9 | 11.9±3.7 |
| Amplitude of early adaptation (Hz) | 62.1±13.3 | 89.8±8.5 | 62.3±16.4 |
| Time constant of early adaptation (ms) | 29.7±4.4 | 28.3±2.2 | 43.1±18.7 |
| Late adaptation (Hz/s) | –11.6±2.3 | –9.6±3.9 | –10.7±1.7 |
| Maximal frequency (Hz) | 23.4±1.7 | 27.6±3.3 | 22.0±2.9 |
| **Action potentials properties** | | | |
| 1$^{st}$ spike amplitude (mV) | 94.8±1.7 | 93.0±5.0 | 88.6±1.9 |
| 1$^{st}$ spike duration (ms) | 1.8±0.1 | 1.9±0.1 | 1.8±0.1 |
| 2$^{nd}$ spike amplitude (mV) | 91.6±1.9 | 91.9±4.5 | 85.5±2.5 |
| 2$^{nd}$ spike duration (ms) | 1.9±0.1 | 1.9±0.1 | 1.9±0.1 |
| Amplitude Reduction (%) | 3.4±0.4 | 1.1±0.7 | 3.6±1.4 |
| Duration Increase (%) | 5.3±0.9 | 3.8±1.5 | 7.5±1.6 |
| **AHP and ADP properties** | | | |
| 1$^{st}$ spike fast AHP (mV) | –8.7±0.8 | –8.7±0.6 | –7.9±1.1 |
| 1$^{st}$ spike ADP (mV) | 0.2±0.1 | 0.2±0.2 | 0±0 |
| 1$^{st}$ spike medium AHP (mV) | –13.8±1.4 | –15.5±0.6 | –13.7±0.8 |
| 1$^{st}$ spike fast AHP latency (ms) | 8.2±1.0 | 7.7±1.4 | 9.8±1.5 |
| 1$^{st}$ spike ADP latency (ms) | 4.6±2.2 | 2.0±2.0 | 0±0 |
| 1$^{st}$ spike, medium AHP latency (ms) | 49.2±3.3 | 48.2±5.4 | 54.8±6.1 |
| 2$^{nd}$ spike fast AHP (mV) | –9.8±1.1 | –8.5±0.5 | –9.6±1.0 |
| 2$^{nd}$ spike ADP (mV) | 0±0 | 0.1±0.1 | 0±0 |
| 2$^{nd}$ spike medium AHP (mV) | –16.1±1.2 | –17.1±0.6 | –15.8±0.4 |
| 2$^{nd}$ spike, fast AHP latency (ms) | 8.6±0.9 | 7.1±0.7 | 11.7±1.3 |
| | F (2.13)=4.063 p=0.0426 | | |
| | * | | |
| | No significant difference in multiple comparisons. | | |

*Table 2 continued on next page*

*Table 2 continued*

| | COXs-negative (n=7) | COX-1 positive (n=4) | COX-2 positive (n=5) |
|---|---|---|---|
| 2nd spike ADP latency (ms) | 1.2±1.2 | 1.7±1.7 | 0±0 |
| 2nd spike, medium AHP latency (ms) | 57.1±5.8 | 51.9±4.6 | 58.3±6.7 |

Data are presented as mean ± SEM. Statistical analyses were performed using a one-way ANOVA (F test) or a Kruskal-Wallis test, depending on the result of the Shapiro-Wilk normality test. If a significant result was found, the corresponding statistics are reported, and post-hoc multiple comparisons are performed.

The online version of this article includes the following source data for table 2:

**Source data 1.** Electrophysiological and molecular properties of pyramidal cells used for *Table 2*.

TTX did not impair the evoked vasoconstriction (*Figure 5—figure supplement 2*). These observations suggest that PGE2 and its EP1 and EP3 receptors mediate a sustained neurogenic vasoconstriction and that once PGE2 is released, its constrictive effect is independent of AP firing.

## Astrocytes through 20-HETE and NPY interneurons are indirect intermediates of pyramidal cell-induced vasoconstriction

In addition to smooth muscle cells, PGE2 released by pyramidal cells can also activate astrocytes and neurons (*Clasadonte et al., 2011*; *Di Cesare et al., 2006*), which also express its receptors (*Tasic et al., 2016*; *Zeisel et al., 2015*). To assess whether astrocytes could mediate the PGE2-dependent vasoconstriction, we first targeted the large conductance $Ca^{2+}$-activated (BK) channels and the 20-HETE pathways, both of which mediate astrocyte-derived vasoconstriction dependent on glutamatergic transmission (*Girouard et al., 2010*; *Mulligan and MacVicar, 2004*). Blockade of BK channels with paxilline (1 µM, *Table 3*, n=10 arterioles) did not impair the vascular response (*Figure 6*; AUC = –3.7 ± 0.3 x $10^3$ %.s, $t_{(18)}$ = 0.03, p=1). Selective inhibition of the 20-HETE synthesizing enzyme, CYP450 $\omega$-hydroxylase, with HET-0016 (100 nM, n=10 arterioles, *Table 3*) reduced the magnitude of the evoked vasoconstriction (*Figure 6*; AUC = –1.6 ± 0.7 x $10^3$ %.s, $t_{(18)}$ = 3.32, p=0.0160). These data suggest that astrocytes partially mediate the vasoconstriction induced by pyramidal cells via 20-HETE

**Table 3.** $IC_{50}$ and concentrations of inhibitors, blocker and antagonists used in tissue.

| Inhibitor/antagonist | In vitro $IC_{50}$ | | Concentrations used for inhibition/antagonism | |
|---|---|---|---|---|
| | | | Preparation | Concentration |
| Indomethacin | COX-1: 22 nM; *Lora et al., 1998* | COX-2: 87 nM | Mouse brain slices; *Lacroix et al., 2015* | 5 µM |
| SC560 | COX-1: 9 nM; *Smith et al., 1998* | COX-2: 6.3 µM | Mouse brain slices; *Lacroix et al., 2015* | 100 nM |
| NS-398 | COX-1: 50 µM; *Lora et al., 1998* | COX-2: 0.6 µM | Mouse brain slices; *Lacroix et al., 2015* | 10 µM |
| ONO-8130 | EP1 receptors: 9.3 nM; *Säfholm et al., 2013a* | | isolated guinea pig trachea; *Säfholm et al., 2013b* | 10 nM |
| L798,106 | EP3 receptors: 0.3 nM (Ki); *Juteau et al., 2001* | | Isolated mouse mesenteric arteries; *Chia et al., 2011* | 1 µM |
| AL8810 | FP receptors: 426 nM (Ki); *Griffin et al., 1999* | | Isolated porcine retinal arterioles; *Oversø Hansen et al., 2015* | 10 µM |
| Paxilline | BK channels: 97 nM; *Tammaro et al., 2004* | | Mouse brain slices; *Girouard et al., 2010* | 1 µM |
| HET-0016 | CYP4A isoforms: 35 nM; *Miyata et al., 2001* | | Mouse brain slices; *Blanco et al., 2008* | 100 nM |
| BIBP3226 | Y1 receptors: 26 nM; *Rudolf et al., 1994* | | Mouse brain slices; *Sun et al., 2003* | 1 µM |

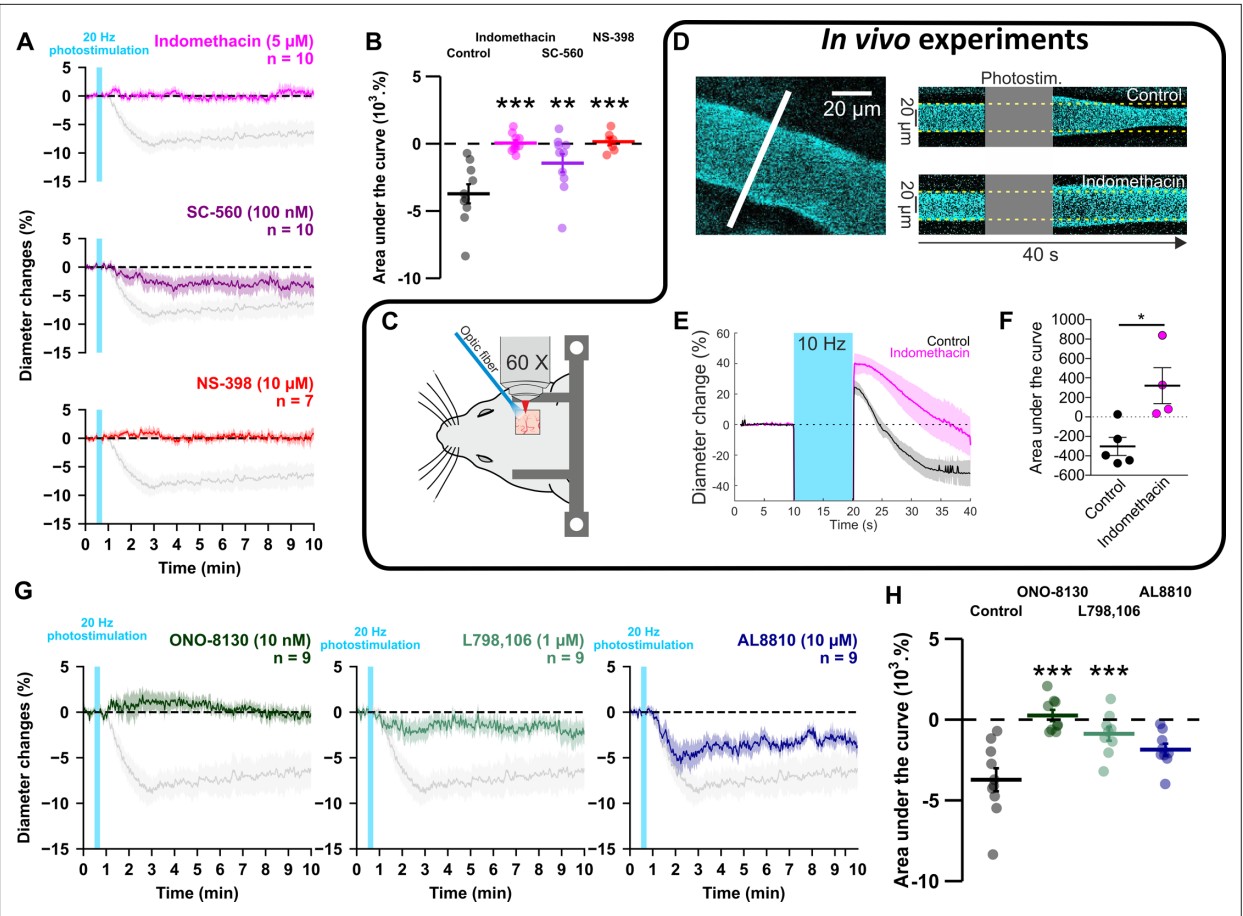

**Figure 5.** GE2 mostly derived from COX-2 activity and its EP1 and EP3 receptors mediates vasoconstriction induced by optogenetically activated pyramidal cells. (**A, B**) Ex vivo effects of the COX1/2 inhibitor indomethacin (magenta, n=10 arterioles from 9 mice), the COX-1 inhibitor SC-560 (purple, n=10 arterioles from 7 mice), and the COX-2 inhibitor NS-398 (red, n=7 arterioles from 6 mice) on kinetics (**A**) and AUC (**B**) of arteriolar vasoconstriction induced by 20 Hz photostimulation (vertical cyan bar). In vivo experiments are highlighted by a black frame. (**C**) Optogenetic stimulation was achieved in vivo with an optic fiber through a chronic cranial window over the barrel cortex. (**D**) Left, diameter of pial arterioles labeled with fluorescein dextran (i.v) was measured with line-scan crossing the vessel (white line). Right, Representative examples of vascular response upon photostimulation (10 Hz, 10 s) under control (top) and indomethacin condition (bottom). (**E**) Diameter changes upon photostimulation under control (black; n=5 arterioles, 4 mice) or indomethacin (magenta; n=4 arterioles, 4 mice) conditions. (**F**) Area under the curve of the diameter change in control (black) or indomethacin (magenta) conditions calculated between 20 and 40 s (unpaired, two-tailed Mann Whitney test, * p<0.05). (**G, H**) Effects of the EP1, EP3 and FP antagonists, ONO-8130 (10 nM, dark green, n=9 arterioles from 7 mice), L798,106 (1 μM, light green, n=9 arterioles from 5 mice) and AL8810 (10 μM, dark blue, n=9 arterioles from 7 mice), respectively, on kinetics (**G**) and AUC (**H**) of arteriolar vasoconstriction induced by 20 Hz photostimulation. The data are shown as the individual values and mean ± SEM. Dashed line represents the baseline. The SEMs envelope the mean traces. The shaded traces in A and G correspond to the control condition (from **Figure 1C** – 20 Hz). *, ** and *** statistically different from 20 Hz control condition with p<0.05, 0.01 and 0.001, respectively.

The online version of this article includes the following source data and figure supplement(s) for figure 5:

**Source data 1.** Diameter measurements (μm) of individual arterioles used to determine diameter changes in **Figure 5A, B, G and H**.

**Source data 2.** Diameter changes of individual arterioles shown in **Figure 5E** under control condition.

**Source data 3.** Diameter changes of individual arterioles shown in **Figure 5E** after indomethacin treatment.

**Figure supplement 1.** Tonic PGE2 does not affect basal vascular tone.

**Figure supplement 1—source data 1.** Diameter measurements (μm) of individual arterioles used to determine diameter changes in **Figure 5—figure supplement 1**.

**Figure supplement 2.** PGE2 dose-dependently induces vasoconstriction.

**Figure supplement 2—source data 1.** Diameter measurements (μm) of individual arterioles used to determine diameter changes in **Figure 5—figure supplement 2**.

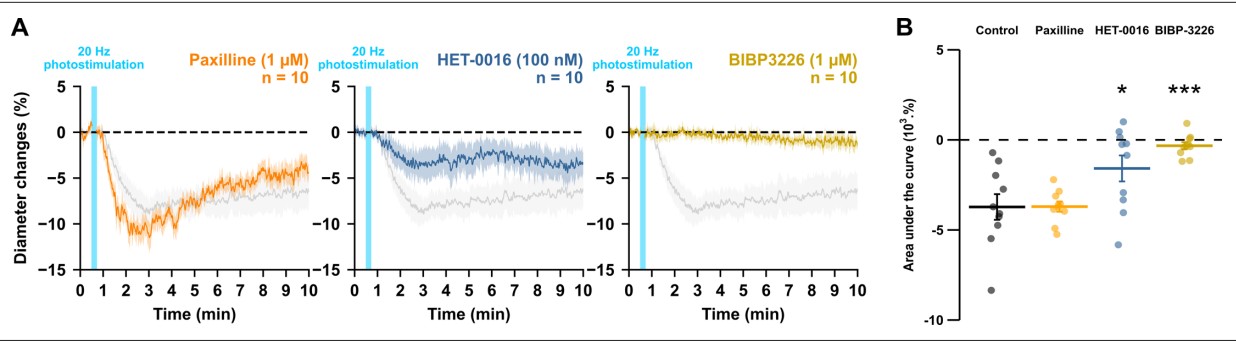

**Figure 6.** NPY Y1 receptors activation and 20-HETE synthesis mediates the vasoconstriction induced by pyramidal neurons. Effects of paxilline (1 μM, orange, n=10 arterioles from 6 mice), HET-0016 (100 nM, blue-grey, n=10 arterioles from 7 mice) and BIBP3226 (1 μM, yellow, n=10 arterioles from 6 mice) on (**A**) kinetics and (**B**) AUC of arteriolar vasoconstriction induced by 20 Hz photostimulation (vertical blue bar). Dashed line represents the baseline. The SEMs envelope the mean traces. The shaded traces correspond to the control condition (**Figure 1C** – 20 Hz). The data are shown as the individual values and mean ± SEM. * and *** statistically different from 20 Hz control condition with p<0.05 and 0.001.

The online version of this article includes the following source data and figure supplement(s) for figure 6:

**Source data 1.** Diameter measurements (μm) of individual arterioles used to determine diameter changes in **Figure 6**.

**Figure supplement 1.** Vasoconstrictive pathways do not influence resting vascular tone.

**Figure supplement 1—source data 1.** Diameter measurements (μm) of individual arterioles used to determine diameter changes in **Figure 6—figure supplement 1**.

but not via K[+] release. We next determined whether NPY, a potent vasoconstrictor (**Cauli et al., 2004**), was involved in neurogenic vasoconstriction. Antagonism of the NPY Y1 receptors by BIBP3226 (1 μM, **Table 3**, n=10 arterioles) abolished neurogenic vasoconstriction (**Figure 6**; AUC = –0.3 ± 0.2 x 10$^3$ %.s, $t_{(18)}$ = 5.28, p=1.9512 x 10$^{-5}$). These results suggest that neurogenic vasoconstriction induced by pyramidal cell photostimulation involves NPY release and the activation of Y1 receptors (**Cauli et al., 2004**; **Karagiannis et al., 2009**; **Uhlirova et al., 2016**) and astrocytes via 20-HETE in a glutamatergic-dependent and -independent manner.

## Discussion

This study establishes that pyramidal cell activity leads to arteriolar vasoconstriction, and that the magnitude of the vasoconstriction depends on AP firing frequency and correlates with a graded increase in pyramidal cell somatic Ca$^{2+}$. This vascular response partially involves glutamatergic transmission through direct and indirect mechanisms on arteriolar smooth muscle cells. Ex vivo and in vivo observations revealed that PGE2, predominantly produced by layer II-III COX-2 pyramidal cells, and its EP1 and EP3 receptors play a crucial role in neurogenic vasoconstriction. Pharmacological evidence indicates that some interneurons, via NPY release and activation of Y1 receptors, and to a lesser extent, astrocytes through 20-HETE and possibly COX-1 derived PGE2 play an intermediary role in this process (**Figure 7**).

We found that increasing the frequency of photostimulation in an ex vivo preparation caused nearby arteriole to go from a barely discernible response to robust vasoconstriction. In contrast, in vivo observations in anesthetized animals, with slower NVC compared to awake animals (**Rungta et al., 2021**; **Uhlirova et al., 2016**), have shown that the optogenetic stimulation of pyramidal cells results in a biphasic response: a fast hyperemic/vasodilatory response (**Kahn et al., 2013**; **Lacroix et al., 2015**; **Scott and Murphy, 2012**), which can be followed by a pronounced vasoconstriction (**Figure 5**; **Uhlirova et al., 2016**). The slow kinetics of the vascular response observed ex vivo is comparable with previous observations in slices (**Cauli et al., 2004**; **Rancillac et al., 2006**), and is likely due to the lower recording temperature compared to in vivo, which slows the synthesis of vasoactive mediators (**Rancillac et al., 2006**) and downstream reactions. The difficulty in observing vasodilation in cortical slices may be due to relaxed arterioles which favor vasoconstriction (**Blanco et al., 2008**). The evidence that neurogenic vasoconstriction is frequency-dependent (**Figure 1**) and that pharmacologically induced vasoconstriction persists in preconstructed (**Girouard et al., 2010**) or pressurized

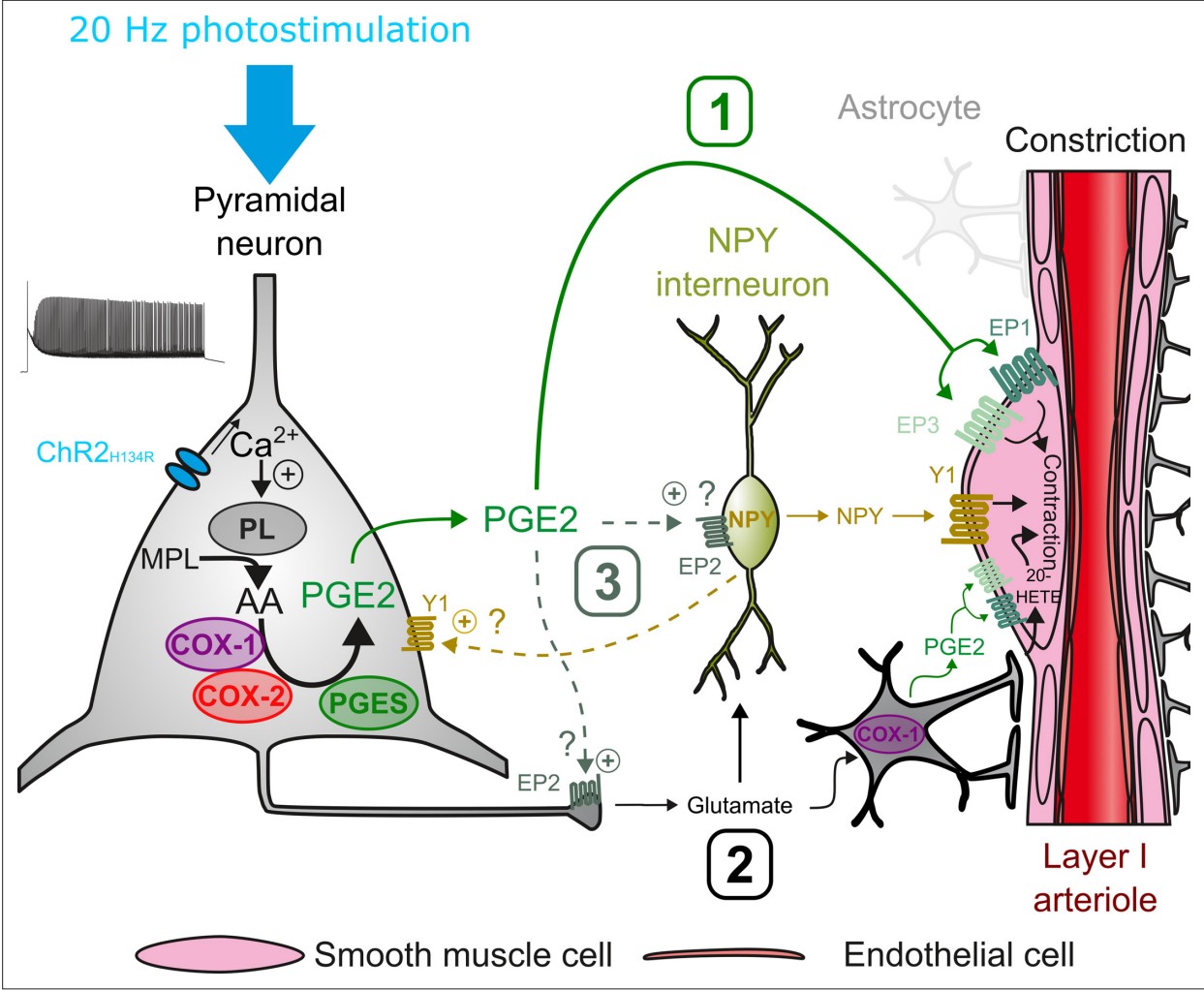

**Figure 7.** Possible pathways of vasoconstriction induced by pyramidal neurons. 20 Hz photostimulation induces activation of pyramidal neurons expressing channelrhodopsin-2 ($ChR2_{H134R}$) and increases intracellular calcium ($Ca^{2+}$). Arachidonic acid (AA) is released from membrane phospholipids (MPL) by phospholipases (PL) activated by intracellular $Ca^{2+}$ and is metabolized by type-1 and type-2 cyclooxygenases (COX-1 and COX-2) and prostaglandin E2 synthases (PGES) to produce prostaglandin E2 (PGE2). Three non-exclusive pathways can be proposed for arteriolar vasoconstriction in layer I: (1) PGE2 released into the extracellular space may act directly on arteriolar EP1 and EP3 receptors to induce smooth muscle cell constriction. (2) Glutamate released from pyramidal cells may activate neuropeptide Y (NPY) interneurons and NPY is released to act on vascular and neuronal Y1 receptors to constrict smooth muscle cells and promote glutamate release, respectively. Glutamate can also activate astrocytes to induce constriction through the 20-HETE and the COX-1/PGE2 pathways. (3) PGE2 may act on pre- and postsynaptic EP2 receptors to facilitate glutamate release and NPY interneuron activation, respectively.

arterioles (*Dabertrand et al., 2013*), suggests that the neurogenic vasoconstriction primarily depends on a high pyramidal cell firing rate rather than on vascular tone.

Most previous observations did not report a decreased in CBF induced by optogenetic stimulation of pyramidal cells in vivo (*Lacroix et al., 2015*; *Scott and Murphy, 2012*). This may be attributed to differences in the photostimulation paradigm and/or the specific subtype of pyramidal cells that were stimulated. In our study, we used 10 s of photostimulation both in vivo and ex vivo. Earlier studies have employed shorter photostimulation times, lasting no more than 1 s (*Lacroix et al., 2015*; *Scott and Murphy, 2012*; *Uhlirova et al., 2016*), which may have resulted in an insufficient number of elicited APs to induce robust vasoconstriction. Furthermore, we observed that neurogenic vasoconstriction is highly dependent on COX-2, which is primarily expressed in layer II-III pyramidal cells (*Lacroix et al., 2015*; *Tasic et al., 2016*; *Zeisel et al., 2015*). In our study, photostimulation of almost all pyramidal cells in Emx1-Cre;Ai32 mice (*Gorski et al., 2002*; *Madisen et al., 2012*) likely resulted in the release of more COX-2 metabolites. Thy1-ChR2 mice used in previous studies (*Scott and Murphy, 2012*;

*Uhlirova et al., 2016*), on the other hand, express primarily ChR2 in layer V pyramidal cells (*Kahn et al., 2013*) which more rarely express COX-2.

Our ex vivo and in vivo observations revealed that PGE2, primarily derived from COX-2, plays a critical role in neurogenic vasoconstriction by activating EP1 and EP3 receptors expressed by vascular smooth muscle cells (*Zhang et al., 2024*). Previous studies have shown that COX-2 pyramidal cells, when activated in vivo by sensory stimulation or ex vivo, induce a NMDA-dependent increase in CBF and vasodilation through PGE2 and EP2/EP4 receptors (*Lacroix et al., 2015*; *Lecrux et al., 2011*; *Niwa et al., 2000*). Differences in the levels and/or sites of action of released PGE2 may explain the absence of secondary vasoconstriction. In Emx1-Cre;Ai32 mice (*Gorski et al., 2002*; *Madisen et al., 2012*), optogenetic stimulation may have activated a greater number of COX-2 pyramidal cells and resulted in a higher local release of PGE2 compared to sensory stimulation. Furthermore, since PGE2 is barely catabolized in the cerebral cortex (*Alix et al., 2008*), most of its removal occurs across the blood-brain barrier by specific transporters. The lack of blood perfusion in brain slices may impair this clearance mechanism, leading to PGE2 accumulation. It is noteworthy that the PGE2-induced vasoconstriction persisted after its removal (*Figure 5—figure supplement 2*). A high level of PGE2 may have facilitated the activation of the EP1 receptor, which has a lower affinity than the EP2/EP4 receptors (*Boie et al., 1997*). Additionally, it may have promoted the rapid desensitization of the dilatory EP4 receptor (*Desai et al., 2000*) thereby favoring vasoconstriction. Furthermore, PGE2 can induce either EP1-dependent arteriolar dilation or constriction depending on whether it is locally applied to capillaries or arterioles. Constriction prevails when both segments are exposed (*Rosehart et al., 2021*). Our photostimulation focused on superficial penetrating arterioles, which lack a capillary network in their close vicinity (*Kasischke et al., 2011*). This may have facilitated the direct EP1-mediated arteriolar constriction (*Dabertrand et al., 2013*; *Rosehart et al., 2021*). Overall, these observations suggest that COX-2 pyramidal cells can sequentially promote both vasodilation and vasoconstriction through the release of PGE2, depending on the context.

Consistent with previous reports in rodents (*Lacroix et al., 2015*; *Tasic et al., 2016*; *Yamagata et al., 1993*; *Zeisel et al., 2015*), the transcripts of the rate-limiting enzymes COX-1 and COX-2, were detected in subpopulations of mouse layer II-III pyramidal cells, respectively. COX-1/2 expression was found to be systematically associated with at least one PGE2 synthesizing enzyme. The major isoforms were cPGES and mPGES2, with the latter being less prevalent (*Lacroix et al., 2015*; *Tasic et al., 2016*; *Zeisel et al., 2015*). The low detection rate of mPGES1, an isoform co-induced with COX-2 by various stimuli (*Takemiya et al., 2007*; *Yamagata et al., 2001*), reflects its low constitutive basal expression level. The presence of PM-PGFS, CBR1 and AKR1B3 in layer II-III pyramidal cells is consistent with single-cell RNAseq data (*Tasic et al., 2016*; *Zeisel et al., 2015*). The expression of a PGFS was systematically observed in COX-2 positive pyramidal cells and in a majority of COX-1 positive neurons, similar to PGES. These observations collectively indicate that subpopulations of layer II-III pyramidal cells express the mRNAs required for PGE2 and PGF2α synthesis derived from COX-1 or COX-2 activity. Our pharmacological observations did not reveal a contribution of PGF2α in neurogenic vasoconstriction, despite the potential ability of pyramidal cells to produce it. This is likely because PGF2α is only detectable in pyramidal neurons under conditions where COX-2 is over-expressed (*Takei et al., 2012*).

Pyramidal cells may have an indirect effect on vascular activity through the activation of intermediate cell types, in addition to the direct vascular effects of PGE2 and glutamate (*Zhang et al., 2024*). Consistent with previous observations, we found that glutamate transmission from pyramidal cells is involved to some extent (*Uhlirova et al., 2016*). Additionally, we found that the NPY Y1 receptor plays a key role in neurogenic vasoconstriction. It is likely that glutamatergic transmission contributed to NPY release, considering that NPY GABAergic interneurons express a wide range of ionotropic and metabotropic glutamate receptors (*Tasic et al., 2016*; *Zeisel et al., 2015*). Consistently, the Y1 receptor has been shown to be involved in vasoconstriction induced by sensory and optogenetic stimulation of GABAergic interneurons (*Uhlirova et al., 2016*). Activation of group I metabotropic receptors in perivascular astrocytes has been shown to promote vasoconstriction via BK channel-dependent $K^+$ release (*Girouard et al., 2010*) or 20-HETE (*Mulligan and MacVicar, 2004*). However, the neurogenic vasoconstriction was not affected by the blockade of BK channels, which rules out this astrocytic pathway. In contrast, the inhibition of $\omega$-hydroxylase partially reduced neurogenic vasoconstriction, suggesting the involvement of 20-HETE. Additionally, astrocytes may also have contributed

to vasoconstriction through the release of PGE2 derived from COX-1 (*Attwell et al., 2016*), as indicated by its mild impairment under SC-560.

The observation that both EP1 and Y1 antagonists abolished the vasoconstriction suggests that PGE2 and NPY may act in series and/or in a more complex manner involving their neuronal receptors. One possibility is that PGE2 activates NPY interneurons via the EP1 receptor. However, NPY interneurons barely express its transcript (*Tasic et al., 2016*; *Zeisel et al., 2015*) and PGE2 constricts arterioles independently of AP firing, suggesting a direct vascular effect of PGE2. Nevertheless, PGE2 may have facilitated NPY release via pre- and postsynaptic EP2-signaling which have been shown to facilitate glutamate release (*Sang et al., 2005*) and to induce neuronal firing (*Clasadonte et al., 2011*), respectively. On the other hand, in addition to smooth muscle cells, Y1 receptors are also enriched in pyramidal neurons (*Smith et al., 2019*), including COX-2 positive ones (*Tasic et al., 2016*), and this receptor has been shown to increase extracellular glutamate in the hippocampus (*Meurs et al., 2012*). By promoting glutamate and possibly PGE2 release, neuronal activation of the Y1 receptor by NPY may also have favored direct (i.e. PGE2) and indirect (i.e. 20-HETE) vasoconstrictive pathways. The combined activation of vascular and neuronal Y1 receptors may explain the complete blockage of optogenetically induced vasoconstriction by its antagonist BIBP3226. In ex vivo relaxed arterioles, where vasoconstriction is favored (*Blanco et al., 2008*), $G_q$ or $G_i$ signaling of EP1 or Y1 receptors, respectively, appears sufficient to induce vasoconstriction. In vivo, where blood flow both induces myogenic tone and allows PGE2 clearance, NPY and PGE2 could also synergistically promote vasoconstriction by decreasing and increasing cAMP and $Ca^{2+}$ levels, respectively, in smooth muscle cells. PGE2 and NPY may also exert temporally distinct vasoconstrictor effects. Indeed, exogenous application of NPY induces a rapid and transient vasoconstriction that returns to baseline levels after removal (*Cauli et al., 2004*), whereas PGE2-induced vasoconstriction is slower and more persistent (*Figure 5—figure supplement 1*). The more transient effect of NPY likely reflects the presence of multiple NPY-degrading enzymes (*Wagner et al., 2015*) and/or the desensitization of the Y1 receptor (*Tsurumaki et al., 2003*; *Gicquiaux et al., 2002*; *Tsurumaki et al., 2002*) which is not the case for PGE2 (*Alix et al., 2008*) and its vasoconstrictor receptors.

In awake mice, synchronous pyramidal cell activity occurs in the absence of any stimulus during the so-called resting state, but it is observed at a much lower frequency than that which triggers vasoconstriction and is associated with increased blood volume (*Ma et al., 2016*). Therefore, neurogenic vasoconstriction described here is unlikely to occur under these conditions. Brief sensory stimulation increases pyramidal cell activity and largely causes vasodilation in both awake and anesthetized animals (*Rungta et al., 2021*). This hyperemic response can be followed by delayed vasoconstriction (*Devor et al., 2007*) and involves NPY/Y1 receptor signaling (*Uhlirova et al., 2016*), similar to the mechanisms reported here. It remains unclear whether PGE2 signaling is also involved in this secondary response. During prolonged sensory stimulation the evoked hyperemic response appears to be more restricted to the activated area at the end of the stimulation than at the beginning (*Berwick et al., 2008*). It is possible that neurogenic vasoconstriction contributes to the later spatial confinement of the vascular response. The time-locked photostimulation of virtually all pyramidal cells leading to vasoconstriction would have resulted in hypersynchrony, a phenomenon that can be observed during sleep/wake transitions (*Asadi-Pooya and Sperling, 2019*). A decrease in hemodynamics has been reported during the transition from rapid eye movement sleep to wakefulness (*Gheres et al., 2023*; *Tsai et al., 2021*), possibly involving neurogenic vasoconstriction. Hypersynchrony is also observed in pathological conditions such as epileptic seizures (*Jiruska et al., 2013*) and in early stages of Alzheimer's disease (*Bezzina et al., 2015*; *Palop et al., 2007*). Although vasoconstriction observed in epilepsy (*Farrell et al., 2016*) exhibits similarities to the neurogenic vasoconstriction described herein, there are notable differences between the two. Like neurogenic vasoconstriction, seizure-induced hypoperfusion is dependent on COX-2 (*Farrell et al., 2016*; *Tran et al., 2020*) and, to some extent on PGE2 (*Farrell et al., 2016*), likely through EP1 and/or EP3 receptors. However, epileptic seizures induce the overexpression of both COX-2 and mPGES1 (*Takemiya et al., 2007*; *Yamagata et al., 1993*) as well as the ectopic expression of NPY (*Baraban, 2004*). Similar transcriptional upregulations have also been reported in Alzheimer's disease (*Bezzina et al., 2015*; *Chaudhry et al., 2008*; *Palop et al., 2007*; *Pasinetti and Aisen, 1998*) Additionally, PGF2α synthesis by COX-2 pyramidal cells is also observed during seizures (*Takei et al., 2012*). Taken together, these observations suggest that the mechanisms governing

neurogenic vasoconstriction are exacerbated in pathological hypersynchrony and may represent potential therapeutic targets.

This neurogenic vasoconstriction, observed during strong pyramidal cell activity, may seem counterintuitive as it would lead to an undersupply of energy substrates despite a high energy demand. However, vasoconstriction has been reported contralateral to the main activated area, despite bilateral increases in neuronal activity and blood glucose uptake (*Devor et al., 2008*), suggesting that neurogenic vasoconstriction plays a physiological role. Through glutamate uptake by astrocytes, neuronal activity stimulates blood glucose uptake and lactate release (*Pellerin and Magistretti, 1994*; *Voutsinos-Porche et al., 2003*). In addition to its role as an oxidative energy substrate for cortical neurons, lactate is also a signaling molecule that enhances their spiking activity (*Karagiannis et al., 2021*). Therefore, uncontrolled lactate supply and metabolism could potentially lead to deleterious hyperactivity (*Cauli et al., 2023*; *Sada et al., 2015*). Thus, the purpose of neurogenic vasoconstriction may be to restrict energy delivery to prevent an overexcitation of the cortical network.

Here, using multidisciplinary approaches, we describe a new mechanism of vasoconstriction that depends on a high firing rate of pyramidal cells. This neurogenic vasoconstriction primarily involves the release of COX-2-derived PGE2 and activation of EP1 and EP3 receptors. It is mediated by direct effects on vascular smooth muscle cells but also by indirect mechanisms involving NPY release from GABAergic interneurons and astrocytes by 20-HETE synthesis. In contrast to previously described mechanisms of neurogenic vasoconstriction, that have been mostly associated with GABAergic interneurons and neuronal inhibition (*Cauli et al., 2004*; *Devor et al., 2007*; *Krawchuk et al., 2020*; *Lee et al., 2021*; *Uhlirova et al., 2016*), our data suggest the involvement of glutamatergic excitatory neurons and increased neuronal activity. This finding will help to update the interpretation of the functional brain imaging signals used to map network activity in health and disease (*Iadecola, 2017*; *Zhang and Raichle, 2010*). This excitatory form of neurogenic vasoconstriction may also help to understand the etiopathogenesis of epilepsy (*Farrell et al., 2016*; *Tran et al., 2020*) and Alzheimer's disease (*Palop and Mucke, 2010*) in which increased cortical network activity and hypoperfusion often overlap.

## Materials and methods
### Animals

Homozygous Emx1-Cre mice Jackson Laboratory, stock #005628, B6.129S2-Emx1$^{tm1(cre)Krj}$/J (*Gorski et al., 2002*) were crossed with homozygous Ai32 mice [Jackson Laboratory, stock #012569, B6;129S-Gt(ROSA)26Sor$^{tm32(CAG-COP4*H134R/EYFP)Hze}$/J (*Madisen et al., 2012*)] to obtain heterozygous Emx1$^{cre/WT}$;Ai32$^{ChR2/WT}$ mice for optogenetic stimulations. C57BL/6RJ mice were used for PGE2 and sulprostone exogenous applications, control optogenetic experiments and single-cell RT-PCR. 16–21 postnatal day-old females and males were used for all ex vivo experiments. Female Emx1$^{cre/WT}$;Ai32$^{ChR2/WT}$ mice, 3- to 5-month-old, were used for in vivo experiments.

All experimental procedures using animals were carried out in strict accordance with French regulations (Code Rural R214/87 to R214/130) and conformed to the ethical guidelines of the European Communities Council Directive of September 22, 2010 (2010/63/UE). Mice were fed ad libitum and housed in a 12 hr light/dark cycle. In vivo experiments were done in accordance with the Institut national de la santé et de la recherche médicale (Inserm) animal care and approved by the ethical committee Charles Darwin (Comité national de réflexion éthique sur l'expérimentation animale – n°5) (protocol number #27135 2020091012114621).

### Ex vivo slice preparation

Mice were deeply anesthetized by isoflurane (IsoVet, Piramal Healthcare UK or IsoFlo, Axience) evaporation in an induction box then euthanized by decapitation. The brain was quickly removed and placed in cold (~4 °C), oxygenated artificial cerebrospinal fluid (aCSF) containing (in mM): 125 NaCl, 2.5 KCl, 1.25 NaH2PO4, 2 CaCl2, 1 MgCl2, 26 NaHCO3, 10 glucose, 15 sucrose and 1 kynurenic acid (Sigma-Aldrich). 300-µm-thick coronal slices containing the barrel cortex were cut with a vibratome (VT1000s; Leica) and were allowed to recover at room temperature for at least 45 min with oxygenated aCSF (95% O2/5% CO2; *Devienne et al., 2018*). The slices were then transferred to a submerged recording

chamber and perfused continuously at room temperature (20–25°C) at a rate of 2 ml/min with oxygenated aCSF lacking kynurenic acid.

## Whole-cell recordings

Patch pipettes (5.5±0.2 MΩ) pulled from borosilicate glass were filled with 8 µl of RNase free internal solution containing (in mM): 144 K-gluconate, 3 MgCl2, 0.5 EGTA, 10 HEPES, pH 7.2 (285/295 mOsm). For electrophysiological recordings combined with calcium imaging, EGTA was replaced by 200 µM Rhod-2 (20777, Cayman chemicals). Whole-cell recordings were performed using a patch-clamp amplifier (Axopatch 200B, MDS). Data were filtered at 5–10 kHz and digitized at 50 kHz using an acquisition board (Digidata 1440, MDS) attached to a personal computer running pCLAMP 10.2 software package (MDS). Electrophysiological properties were determined in current-clamp mode (*Karagiannis et al., 2009*). Membrane potential values were corrected for theoretical liquid junction potential (−15.6 mV). Resting membrane potential of neurons was measured immediately after passing in whole-cell configuration. Only neurons with a resting membrane potential more hyperpolarized than −60 mV were analyzed further.

## Optogenetic stimulation

Optogenetic stimulation was achieved through the objective using a 470 nm light emitting device (LED, CoolLED, Precise Excite) attached to the epifluorescence port of a BX51WI microscope (Olympus) and a set of multiband filters consisting of an excitation filter (HC 392/474/554/635, Semrock), a dichroic mirror (BS 409/493/573/652, Semrock), and an emission filter (HC 432/515/595/730, Semrock). Photostimulation consisted of a 10 s train of 5ms light pulses at an intensity of 38 mW/mm² and delivered at five different frequencies (1, 2, 5, 10, and 20 Hz).

## Infrared imaging

Blood vessels and cells were observed in slices under infrared illumination with Dodt gradient contrast optics (IR-DGC, Luigs and Neumann) using a double-port upright microscope (BX51WI, Olympus) and a collimated light emitting device (LED; 780 nm; ThorLabs) as the transmitted light source, a 40 X (LUMPlanF/IR, 40 X/0.80 W, Olympus) or a 60 X (LUMPlan FL/IR 60 X/0.90 W, Olympus) objective and a digital camera (OrcaFlash 4.0, Hamamatsu) attached to the front port of the microscope. Penetrating arterioles in layer I were selected by IR-DGC videomicroscopy based on their well-defined luminal diameter (10–40 µm), their length remaining in the focal plane for at least 50 µm (*Lacroix et al., 2015*), and the thickness of their wall (4.1±0.1 µm, n=176 blood vessels). A resting period of at least 30 min (*Zonta et al., 2003*) was observed after slice transfer. After light-induced responses, arteriolar contractility was tested by the application of aCSF containing the thromboxane A2 agonist, U46619 (100 nM; *Cauli et al., 2004*) or K⁺ enriched solution (composition in mM: 77.5 NaCl, 50 KCl, 1.25 NaH2PO4, 2 CaCl2, 1 MgCl2, 26 NaHCO3, 10 glucose, 15 sucrose). Vessels that did not constrict with these applications were discarded. Only one arteriole was monitored per slice receiving a single optogenetic or pharmacological stimulation. IR-DGC images were acquired at 0.1 Hz for pharmacological applications and at 1 Hz for optogenetic experiments using Imaging Workbench 6.1 software(Indec Biosystems). The focal plane was continuously maintained on-line using IR-DGC images of cells as anatomical landmarks (*Lacroix et al., 2015*).

## Calcium imaging

Visually and electrophysiologically identified layer II-III pyramidal cells were filled with the calcium-sensitive dye Rhod-2 (200 µM, Cayman chemical, 20777) using patch pipettes. Optical recordings were made at least 15 min after passing in whole-cell configuration to allow for somatic diffusion of the dye. Rhod-2 was excited with a 585 nm LED (Cool LED, Precise Excite) at an intensity of 0.56 mW/mm² and the filter set used for optogenetic stimulation using the Imaging Workbench 6.1 software(Indec Byosystems). IR-DGC and fluorescence images were acquired by alternating epifluorescence and transmitted light sources. IR-DGC and fluorescence were respectively sampled at 5 Hz and 1 Hz during baseline and optogenetic stimulation, respectively, and at 1 Hz and 0.2 Hz after photostimulation. During photostimulation, bleed-through occurred in the Rhod-2 channel due to the fluorescence of the EYFP-ChR2 transgene (*Madisen et al., 2012*). Therefore, the Ca²⁺ response could not be reliably analyzed during this period. To compensate for potential x-y drifts, all images were

registered off-line using the 'StackReg' plug-in (*Thévenaz et al., 1998*) of the ImageJ 1.53 software. To define somatic regions of interest (ROIs), the soma was manually delineated from IR-DGC images. Fluorescence intensity changes ($\Delta F/F_0$) were expressed as the ratio $(F-F_0)/F_0$ where F is the mean fluorescence intensity in the ROI at a given time point, and $F_0$ is the mean fluorescence intensity in the same ROI during the 30 s control baseline.

## Drugs

All pharmacological compounds were bath applied after a 5-min baseline, and vascular dynamics were recorded during bath application. The following drugs were dissolved in water: D-(-)–2-amino-5-phosphonopentanoic acid (D-AP5, 50 µM, Hello Bio, HB0225), 6,7-dinitroquinoxaline-2,3-dione (DNQX, 10 µM, Hello Bio, HB0262), LY367385 (100 µM, Hello Bio, HB0398) and BIBP3226 (1 µM, Tocris, 2707). Tetrodotoxin (TTX, 1 µM, L8503, Latoxan) was dissolved in 90% acetic acid. PGE2 (HB3460, Hello Bio), sulprostone (10 µM, Cayman chemical, 14765), 2-methyl-6-(phenylethynyl)pyridine (MPEP, 50 µM, Hello Bio, HB0426) and 9,11-dideoxy-9α,11α-methanoepoxy prostaglandin F2α (U-46619, 100 nM, Enzo, BML-PG023) were dissolved in ethanol. Indomethacin (5 µM, Sigma-Aldrich, I7378), SC-560 (100 nM, Sigma-Aldrich, S2064-5MG), NS-398 (10 µM, Enzo, BML-EI261), ONO-8130 (10 nM, Tocris, 5406), L-798,106 (1 µM, Cayman chemical, 11129), AL8810 (10 µM, Cayman chemical, 16735), paxilline (10 µM, Tocris, 2006) and HET0016 (100 µM, Merck, SML2416-5MG) in DMSO. Acetic acid, ethanol and DMSO doses were always used below 0.1%. Synthesis inhibitors and BIBP3226 were applied at least 30 min before optogenetic stimulation. TTX was applied at least 15 min before, while glutamate receptor antagonists and paxilline were applied at least 10 and 5 min before, respectively.

## Vascular reactivity analysis

To compensate for potential x-y drifts, all images were realigned off-line using the 'StackReg' plug-in *Thévenaz et al., 1998* of the ImageJ 1.53 software. Luminal diameter was measured in layer I on registered images using custom analysis software developed in MATLAB (MathWorks) (*Lacroix et al., 2015*). To avoid potential drawbacks due to vessel instability, only arterioles with a stable diameter were analyzed further. Arterioles were considered stable if the relative standard deviation of their diameter during the baseline period was less than 5% (*Lacroix et al., 2015*). Comparison of the mean arteriolar diameter during the 5 min baseline and the 5 min final pharmacological treatments revealed that all drugs, except AL8810, had no effect on resting diameter (*Figure 3—figure supplement 1*, *Figure 5—figure supplement 1*, *Figure 6—figure supplement 1*).

Diameter changes ($\Delta D/D_0$) were expressed as $(D_t – D_0)/D_0$ where $D_t$ is the diameter at the time t and $D_0$ is the mean diameter during the baseline period. To eliminate sharp artifacts due to transient loss of focus, diameter change traces were smoothed using a sliding three-point median filter. The overall vascular response over time was captured by the area under the curve of diameter changes after photostimulation. To determine the onset of vasoconstriction, a Z-score was calculated from the diameter change traces using the formula: $Z = (x - \mu)/\sigma$, where both the mean µ and the standard deviation σ were calculated from the values before photostimulation. Onset of vasoconstriction was defined as the time after the start of photostimulation at which the Z-score exceeded or fell below -a value of –1.96 (95% criteria) for 10 s. If a vessel showed no vasoconstriction, the onset was arbitrarily set at 1800s. Graphs were generated using R software version 4.3.0 (*R Development Core Team, 2023*) and Matplotlib package (*Caswell, 2023*).

## Intrinsic optical signals analysis

Variations in IR light transmittance (ΔT) (*Zhou et al., 2010*) were determined using ImageJ 1.53 software according to: $\Delta T = (T_t - T_0)/T_0$ where $T_t$ is the light transmittance at a time t and T0 is the average light transmittance during the baseline period of a squared region of interest of 100 µm x 100 µm manually delineated in layer I. The rate of ΔT change was determined as the first derivative of ΔT (dΔT/dt, where ΔT is the change in light transmittance and t is time). Slices that showed a maximum rate of increase of dΔT/dt greater than 2% /s, indicating the occurrence of spreading depression (*Zhou et al., 2010*), were excluded.

## Surgery

Chronic cranial windows were implanted one week after the head bar surgery as previously described (**Tournissac et al., 2022**). We used a 100 μm thick glass coverslip over the barrel cortex (~4 mm$^2$). Before two-photon experiments, a recovery period of 7–10 days minimum was respected.

## In vivo two-photon imaging and photostimulation

For two-photon excitation, we used a femtosecond laser (Mai Tai eHP; SpectraPhysics) with a dispersion compensation module (Deepsee; SpectraPhysics) emitting 70-fs pulses at 80 MHz. The laser power was attenuated by an acousto-optical modulator (AA Optoelectronic, MT110-B50-A1.5-IR-Hk). Scanning was performed with Galvanometric scanner (GS) mirrors (8315KM60B; Cambridge Technology). Fluorescein was excited at 920 nm and the emitted light was collected with a LUMFLN60XW (Olympus, 1.1 NA) water immersion objective. Collected photons were sorted using a dichroic mirror centered at 570 nm, a FF01-525/50 nm filter (Semrock) and a GaAsP (Hamamatsu) photomultipliers tube. Customized LabView software was used to control the system. Line-scans were drawn across pial vessels to measure the change in arterioles diameter, which are not compromised by the fluorescence from the ChR2-EYFP transgene in the parenchyma (**Madisen et al., 2012**), are less affected by potential movement in the x-y plan than in penetrating arterioles, and whose dilation dynamics are similar in the somatosensory cortex (**Rungta et al., 2021**).

Mice were anesthetized with a mixture of ketamine and medetomidine (100 and 0.5 mg/kg, respectively, intraperitoneally (i.p.)) during imaging sessions. Body temperature was maintained at 36.5°C with a retro-controlled heating pad. Fluorescein dextran (70 kDa) was injected i.v. through a retro-orbital injection to label brain vessels. Mice received continuous air through a nose cone supplemented with oxygen to reach a final concentration of 30% $O_2$.

Photostimulation was delivered with a 473 nm laser (Coblot MLD, Sweden) through an optical fiber placed above the glass coverslip and directed at the pial artery of interest. Each photostimulation consisted of a 10-second train of 5 ms light pulses at an intensity of 1 mW delivered at 10 Hz, with a 5-minute interstimulus interval to allow full recovery to baseline. Indomethacin (10 mg/kg, #15425529, Thermo Fisher Scientific) was administered i.v. through a retro-orbital injection.

## Imaging analysis

Pial arteriole diameter change was determined with line-scan acquisitions and a home-made Matlab script as previously described (**Rungta et al., 2018**). Trials from the same vessel were averaged (with a 0.1 s interpolation) for analysis. Area under the curve and statistics were performed using GraphPad Prim (version 6).

## Cytoplasm harvesting and single-cell RT-PCR

At the end of the whole-cell recording, which lasted less than 15 min, the cytoplasmic content was collected in the recording pipette by applying a gentle negative pressure. The pipette's content was expelled into a test tube and RT was performed in a final volume of 10 μl as described previously (**Devienne et al., 2018**). The scRT-PCR protocol was designed to probed simultaneously the expression of prostaglandins synthesizing enzymes and neuronal markers (**Lacroix et al., 2015**). Prostaglandins synthesizing enzymes included COX-1 and COX-2, the terminal PGE2 synthases (PGES): mPGES1, mPGES2 and cPGES, the terminal PGF2αsynthases (PGFS): PM-PGFS (Prxl2b) and AKR1B3 and the carbonyl reductase CBR1. Neuronal markers included the vesicular glutamate transporter, vGluT1, and the two isoforms of glutamic acid decarboxylase, GAD65 and GAD67. Two-step amplification was performed essentially as described (**Devienne et al., 2018**). First, cDNAs present in the 10 μl reverse transcription reaction were simultaneously amplified with all external primer pairs listed in Appendix 1—key resources table. Taq polymerase (2.5 U; QIAGEN) and external primers mix (20 pmol each) were added to the manufacturer's buffer (final volume, 100 μl), and 20 cycles (95 ∘ C, 30 s; 60 ∘ C, 30 s; and 72 ∘ C, 35 s) of PCR were performed. Second rounds of PCR were performed using 1 μl of the first PCR product as a template. In this second round, each cDNA was amplified individually using its specific nested primer pair (Appendix 1—key resources table) by performing 35 PCR cycles (as described above). 10 μl of each individual PCR product were run on a 2% agarose gel stained with ethidium bromide using ΦX174 digested by *HaeIII* as a molecular weight marker. The efficiency of the protocol was validated using 500 pg of total forebrain RNAs (**Figure 4—figure supplement 1**).

## Statistical analyses

Statistical analyses were performed using GraphPad Prism version 7.00 for Windows (GraphPad Software, La Jolla California USA, https://www.graphpad.com/) and R software version 4.3.0 (*R Development Core Team, 2023*). Normality of distribution was assessed using the Shapiro-Wilk tests. Equality of variance was assessed using Brown-Forsythe tests for comparisons between groups and using F-tests for comparisons with a control group. Parametric tests were only used if these criteria were met. Statistical significance of morphological and physiological properties of penetrating arterioles was determined using one-way ANOVA for comparison between groups. Statistical significance of calcium was determined using two-tailed unpaired t-tests and Statistical significance of vascular responses were appreciated using Tukey posthoc tests for the different frequencies conditions and using Dunnett's posthoc tests for the different pharmacological conditions compared to the 20 Hz condition without pharmacological compound. False discovery rate correction was used for multiple comparisons. Statistical significance of vascular diameter for drug applications was determined using two-tailed paired t-tests. Statistical significance on all figures uses the following convention: *$p<0.05$, **$p<0.01$ and ***$p<0.001$.

## Acknowledgements

The authors thank Dr Rebecca Piskorowski for constructive criticism of the manuscript. We acknowledge the invaluable support of the animal facilities of IBPS (RongIBPS) and Institut de la vision for their expert care and maintenance of the animals used in this study. Financial support was provided by grants from the Agence Nationale pour la Recherche (ANR-17-CE37-0010-03, BC; CE37_2020_TF-fUS-CADASIL, SC; ANR-20-CE14-0025, DL; ANR-23-CE14-0038-01, BC), the Fondation Alzheimer France (M21JRCN009, SC) and the i-Bio initiative of Sorbonne University (BC). BLG and EB were supported by fellowships from Fondation pour la Recherche sur Alzheimer and MT by a fellowship from the Fondation pour la Recherche Médicale (SPF201909009103)

## Additional information

### Funding

| Funder | Grant reference number | Author |
|---|---|---|
| Fondation pour la Recherche sur Alzheimer | | Benjamin Le Gac Esther Belzic |
| Agence Nationale de la Recherche | ANR-17-CE37-0010-03 | Bruno Cauli |
| Agence Nationale de la Recherche | CE37_2020_TF-fUS-CADASIL | Serge Charpak |
| Agence Nationale de la Recherche | ANR-20-CE14-0025 | Dongdong Li |
| Agence Nationale de la Recherche | ANR-23-CE14-0038-01 | Bruno Cauli |
| FONDATION ALZHEIMER | M21JRCN009 | Serge Charpak |
| i-Bio Initiative | | Bruno Cauli |
| Fondation pour la Recherche Médicale | SPF201909009103 | Marine Tournissac |

The funders had no role in study design, data collection and interpretation, or the decision to submit the work for publication.

### Author contributions

Benjamin Le Gac, Conceptualization, Data curation, Formal analysis, Funding acquisition, Investigation, Visualization, Methodology, Writing – original draft, Writing – review and editing; Marine Tournissac, Conceptualization, Data curation, Formal analysis, Investigation, Visualization, Methodology,

Writing – review and editing; Esther Belzic, Data curation, Formal analysis, Funding acquisition, Investigation, Writing – review and editing; Sandrine Picaud, Formal analysis, Investigation; Isabelle Dusart, Writing – review and editing; Hédi Soula, Data curation; Dongdong Li, Conceptualization, Data curation, Funding acquisition, Investigation, Methodology, Writing – review and editing; Serge Charpak, Conceptualization, Resources, Data curation, Formal analysis, Supervision, Funding acquisition, Investigation, Methodology, Writing – review and editing; Bruno Cauli, Conceptualization, Resources, Data curation, Formal analysis, Supervision, Funding acquisition, Validation, Investigation, Visualization, Methodology, Writing – original draft, Project administration, Writing – review and editing

### Author ORCIDs
Benjamin Le Gac ⓘ https://orcid.org/0000-0002-4703-6217
Marine Tournissac ⓘ https://orcid.org/0000-0002-1990-5075
Isabelle Dusart ⓘ https://orcid.org/0000-0002-3211-4323
Hédi Soula ⓘ https://orcid.org/0000-0001-5306-9712
Dongdong Li ⓘ https://orcid.org/0000-0002-6731-4771
Serge Charpak ⓘ https://orcid.org/0000-0002-5516-1245
Bruno Cauli ⓘ https://orcid.org/0000-0003-1471-4621

### Ethics
All experimental procedures using animals were carried out in strict accordance with French regulations (Code Rural R214/87 to R214/130) and conformed to the ethical guidelines of the European Communities Council Directive of September 22, 2010 (2010/63/UE). Mice were fed ad libitum and housed in a 12-hour light/dark cycle. In vivo experiments were done in accordance with the Institut national de la santé et de la recherche médicale (Inserm) animal care and approved by the ethical committee Charles Darwin (Comité national de réflexion éthique sur l'expérimentation animale - n°5) (protocol number #27135 2020091012114621).

Reviewer #1 (Public review): https://doi.org/10.7554/eLife.102424.3.sa1
Reviewer #2 (Public review): https://doi.org/10.7554/eLife.102424.3.sa2
Author response https://doi.org/10.7554/eLife.102424.3.sa3

## Additional files

### Supplementary files
MDAR checklist

Source code 1. Matlab script for blood vessel analysis. The method has been described in *Lacroix et al., 2015*.

### Data availability
All data generated or analyzed during this study are included in the manuscript and supporting files. Source data files have been provided for Figures 1 to 6. Source code for blood vessel analysis is provided in *Source code 1*.

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

# Appendix 1

## Appendix 1—key resources table

| Reagent type (species) or resource | Designation | Source or reference | Identifiers | Additional information |
|---|---|---|---|---|
| Strain, strain background (*Mus musculus*, male and female) | C57BL/6RJ, Wild type | Janvier Labs | C57BL/6RJ | |
| Strain, strain background (*Mus musculus*, male and female) | B6.129P2- *Emx1*[tm1(cre)Krj]/J, Emx1 [Cre/Cre] | PMID:12151506; *Gorski et al., 2002* | RRID:IMSR_JAX:005628 | |
| Strain, strain background (*Mus musculus*, male and female) | B6.129P2- Gt(ROSA)26Sor[tm32(CAG-COP4*H134R/EYFP)Hze], Gt(ROSA)26Sor [ChR2(H134R)-EYFP/ ChR2(H134R)-EYFP] | PMID:22446880; *Madisen et al., 2012* | RRID:IMSR_JAX:024109 | |
| Sequence-based reagent | *Slc17a7* external sense PCR primer (vGluT1) | PMID:23565079; *Cabezas et al., 2013* | | GGCTCCTTTTTCTGGGGCTAC |
| Sequence-based reagent | *Slc17a7* external antisense PCR primer (vGluT1) | PMID:23565079; *Cabezas et al., 2013* | | CCAGCCGACTCCGTTCTAAG |
| Sequence-based reagent | *Slc17a7* internal sense PCR primer (vGluT1) | PMID:23565079; *Cabezas et al., 2013* | | ATTCGCAGCCAACAGGGTCT |
| Sequence-based reagent | *Slc17a7* internal antisense PCR primer (vGluT1) | PMID:23565079; *Cabezas et al., 2013* | | TGGCAAGCAGGGTATGTGAC |
| Sequence-based reagent | *Gad2* external sense PCR primer (GAD 65) | PMID:19295167; *Karagiannis et al., 2009* | | CCAAAAGTTCACGGGCGG |
| Sequence-based reagent | *Gad2* external antisense PCR primer (GAD 65) | PMID:19295167; *Karagiannis et al., 2009* | | TCCTCCAGATTTTGCGGTTG |
| Sequence-based reagent | *Gad2* internal sense PCR primer (GAD 65) | PMID:22754499; *Perrenoud et al., 2012* | | CACCTGCGACCAAAAACCCT |
| Sequence-based reagent | *Gad2* internal antisense PCR primer (GAD 65) | PMID:22754499; *Perrenoud et al., 2012* | | GATTTTGCGGTTGGTCTGCC |
| Sequence-based reagent | *Gad1* external sense PCR primer (GAD 67) | PMID:12196560; *Férézou et al., 2002* | | TACGGGGTTCGCACAGGTC |
| Sequence-based reagent | *Gad1* external antisense PCR primer (GAD 67) | PMID:12196560; *Cabezas et al., 2013* | | CCCAGGCAGCATCCACAT |
| Sequence-based reagent | *Gad1* internal sense PCR primer (GAD 67) | PMID:23565079; *Cabezas et al., 2013* | | CCCAGAAGTGAAGACAAAAGGC |
| Sequence-based reagent | *Gad1* internal antisense PCR primer (GAD 67) | PMID:23565079; *Cabezas et al., 2013* | | AATGCTCCGTAAACAGTCGTGC |
| Sequence-based reagent | *Ptgs1* external sense PCR primer (COX-1) | This paper | | ATCCCTGTTGTTACTATCCGTGC |
| Sequence-based reagent | *Ptgs1* external antisense PCR primer (COX-1) | This paper | | TGTGGGGCAGTCTTTGGGTA |
| Sequence-based reagent | *Ptgs1* internal sense PCR primer (COX-1) | This paper | | AGGGTGTCTGTGTCCGCTTT |

*Appendix 1 Continued on next page*

*Appendix 1 Continued*

| Reagent type (species) or resource | Designation | Source or reference | Identifiers | Additional information |
|---|---|---|---|---|
| Sequence-based reagent | *Ptgs1* internal antisense PCR primer (COX-1) | This paper | | GGCTGGGGATAAGGTTGGAC |
| Sequence-based reagent | *Ptgs2* external sense PCR primer (COX-2) | PMID:21734275; *Lecrux et al., 2011* | | CTGAAGCCCACCCCAAACAC |
| Sequence-based reagent | *Ptgs2* external antisense PCR primer (COX-2) | PMID:29985318; *Devienne et al., 2018* | | CCTTATTTCCCTTCACACCCAT |
| Sequence-based reagent | *Ptgs2* internal sense PCR primer (COX-2) | PMID:29985318; *Devienne et al., 2018* | | AACAACATCCCCTTCCTGCG |
| Sequence-based reagent | *Ptgs2* internal antisense PCR primer (COX-2) | PMID:29985318; *Devienne et al., 2018* | | TGGGAGTTGGGCAGTCATCT |
| Sequence-based reagent | *Ptges* external sense PCR primer (mPGES1) | This paper | | GCCTGGTGATGGAGAGCG |
| Sequence-based reagent | *Ptges* external antisense PCR primer (mPGES1) | This paper | | GGAGCGAAGGCGTGGGTT |
| Sequence-based reagent | *Ptges* internal sense PCR primer (mPGES1) | This paper | | AGATGAGGCTGCGGAAGAAG |
| Sequence-based reagent | *Ptges* internal antisense PCR primer (mPGES1) | This paper | | CACGAAGCCGAGGAAGAGGA |
| Sequence-based reagent | *Ptges2* external sense PCR primer (mPGES1) | This paper | | CGACTTCCACTCCCTGCC |
| Sequence-based reagent | *Ptges2* external antisense PCR primer (mPGES2) | This paper | | CATCTCCTCCGTCCTGGCTT |
| Sequence-based reagent | *Ptges2* internal sense PCR primer (mPGES2) | This paper | | GAGGTGAATCCCGTGAGAAGG |
| Sequence-based reagent | *Ptges2* internal antisense PCR primer (mPGES2) | This paper | | TTCCTTCCCGCCATACATCT |
| Sequence-based reagent | *Ptges3* external sense PCR primer (cPGES) | This paper | | TCCAAGCATAAAAGAACAGACAGA |
| Sequence-based reagent | *Ptges3* external antisense PCR primer (cPGES) | This paper | | TGGCATCTTTTCATCATCACTGTC |
| Sequence-based reagent | *Ptges3* internal sense PCR primer (cPGES) | This paper | | TAACAAAGGAAAGGGCAAAGC |
| Sequence-based reagent | *Ptges3* internal antisense PCR primer (cPGES) | This paper | | CATCATCTGCTCCATCTACTTCTG |
| Sequence-based reagent | *Prxl2b* external sense PCR primer (PM-PGFS) | This paper | | AGGAGTTTCTGGATGGTGGTTAC |
| Sequence-based reagent | *Prxl2b* external antisense PCR primer (PM-PGFS) | This paper | | CACCTCCCACACACCTCTTCAT |
| Sequence-based reagent | *Prxl2b* internal sense PCR primer (PM-PGFS) | This paper | | ACCTGTTCGTGATGTAGCCTCC |
| Sequence-based reagent | *Prxl2b* internal antisense PCR primer (PM-PGFS) | This paper | | CTGGGGTGGCTTGCTGGA |
| Sequence-based reagent | *Akr1b1* external sense PCR primer (Akr1b3) | This paper | | CAGAATGAGAAGGAGGTGGGA |
| Sequence-based reagent | *Akr1b1* external antisense PCR primer (Akr1b3) | This paper | | TTGAAGTTGGAGACACCGATTG |

*Appendix 1 Continued on next page*

*Appendix 1 Continued*

| Reagent type (species) or resource | Designation | Source or reference | Identifiers | Additional information |
|---|---|---|---|---|
| Sequence-based reagent | *Akr1b1* internal sense PCR primer (Akr1b3) | This paper | | CAAGGAGCAGGTGGTGAAGC |
| Sequence-based reagent | *Akr1b1* internal antisense PCR primer (Akr1b3) | This paper | | CATAGCCGTCCAAGTGTCCA |
| Sequence-based reagent | *Cbr1* external sense PCR primer (CBR1) | This paper | | AACCCGCAGAGCATTCGC |
| Sequence-based reagent | *Cbr1* external antisense PCR primer (CBR1) | This paper | | GCCAACCTTCTTCCGCAT |
| Sequence-based reagent | *Cbr1* internal sense PCR primer (CBR1) | This paper | | CAATGACGACACCCCCTTCC |
| Sequence-based reagent | *Cbr1* internal antisense PCR primer (CBR1) | This paper | | CTCCTCTGTGATGGTCTCGCTT |
| Chemical compound, drug | Rhod-2 | Cayman chemical | 20777 | |
| Chemical compound, drug | 9,11-dideoxy-9α,11α-methanoepoxy prostaglandin F2α | Enzo | BML-PG023 | |
| Chemical compound, drug | Tetrodotoxin | Latoxan | L8503 | |
| Chemical compound, drug | D-(-)–2-amino-5-phosphonopentanoic acid | Hello Bio | HB0225 | |
| Chemical compound, drug | 6,7-dinitroquinoxaline-2,3-dione | Hello Bio | HB0262 | |
| Chemical compound, drug | LY367385 | Hello Bio | HB0398 | |
| Chemical compound, drug | 2-methyl-6-(phenylethynyl)pyridine | Hello Bio | HB0426 | |
| Chemical compound, drug | Indomethacin | Sigma-Aldrich | I7378 | |
| Chemical compound, drug | SC-560 | Sigma-Aldrich | S2064 | |
| Chemical compound, drug | NS-398 | Enzo | BML-EI261 | |
| Chemical compound, drug | ONO-8130 | Tocris | 5406 | |
| Chemical compound, drug | L-798,106 | Cayman chemical | 11129 | |
| Chemical compound, drug | AL8810 | Cayman chemical | 16735 | |
| Chemical compound, drug | PGE2 | Hello Bio | HB3460 | |
| Chemical compound, drug | Sulprostone | Cayman chemical | 14765 | |
| Chemical compound, drug | BIBP3226 | Tocris | 2707 | |
| Chemical compound, drug | HET0016 | Merck | SML2416 | |
| Chemical compound, drug | paxilline | Tocris | 2006 | |

*Appendix 1 Continued on next page*

*Appendix 1 Continued*

| Reagent type (species) or resource | Designation | Source or reference | Identifiers | Additional information |
|---|---|---|---|---|
| Chemical compound, drug | Dithiothreitol | VWR | 443852 A | |
| Chemical compound, drug | Primer "random" | Roche | 11034731001 | |
| Chemical compound, drug | dNTPs | GE Healthcare Life Sciences | 28-4065-52 | |
| Chemical compound, drug | Mineral Oil | Sigma-Aldrich | M5904 | |
| Chemical compound, drug | RNasin Ribonuclease Inhibitors | Promega | N2511 | |
| Chemical compound, drug | SuperScript II Reverse Transcriptase | Invitrogen | 18064014 | |
| Chemical compound, drug | Taq DNA Polymerase | Qiagen | 201205 | |
| Software, algorithm | Pclamp v 10.2 | Molecular Devices | RRID:SCR_011323 | |
| Software, algorithm | Matlab v 2018b | MathWorks | RRID:SCR_001622 | |
| Software, algorithm | GraphPad Prism v 7 | GraphPad | RRID:SCR_002798 | |
| Software, algorithm | ImagingWorkbench v 6.1 | INDEC Systems | | |
| Software, algorithm | FIJI | PMID:22743772; *Schindelin et al., 2012* | RRID:SCR_002285 | |
| Software, algorithm | R v 4.3.0 | R Core Team | RRID:SCR_001905 | |
| Other | Vibratome | Leica | VT1000S RRID:SCR_016495 | |
| Other | Upright microscope | Olympus | BX51WI | |
| Other | Dual port module | Olympus | WI-DPMC | |
| Other | 60 x Objective | Olympus | LUMPlan Fl /IR 60 x/0.90 W | |
| Other | 40 x Objetive | Olympus | LUMPlan Fl /IR 40 x/0.80 W | |
| Other | sCMOS camera | Hamamatsu | ORCA-Flash4.0 | |
| Other | Axopatch 200B | Molecular Devices | RRID:SCR_018866 | |
| Other | Digidata 1440 A | Molecular Devices | RRID:SCR_021038 | |
| Other | S900 stimulator | Dagan corporation | | |
| Other | pE-2 | CoolLED | | |
| Other | Excitation filter | Semrock | HC 392/474/554/635 | |
| Other | Dichroic mirror | Semrock | BS 409/493/573/652 | |
| Other | Emission filter | Semrock | HC 432/515/595/730 | |
| Other | 780 nm Collimated LED | Thorlabs | M780L3-C1 | |
| Other | Dodt Gradient Contrast | Luigs and Neumann | 200–100 200 0155 | |
| Other | Beam splitter | Semrock | 725 DCSPXR | |
| Other | Analogic CCD camera | Sony | XC ST-70 CE | |

