## [Editor Report · eLife Assessment]

This study presents **important** findings on the role of pyramidal cells driving vasoconstriction in brain arteries through a COX-2/PGE2 pathway, with additional contributions from NPY (interneurons) and 20-HETE (astrocytes). Optogenetic stimulation of cortical pyramidal neurons induces vasoconstriction, potentially leading to oxygen and nutrient undersupply in regions with sustained activation - a mechanism potentially relevant under pathological conditions. The authors provide **convincing** evidence from brain slice experiments and some in vivo data from anesthetized animals, carefully discussing the strengths and limitations of both approaches.

---

## [Referee Report · Reviewer #1 (Public review)]

SNeuronal activity spatiotemporal fine-tuning of cerebral blood flow balances metabolic demands of changing neuronal activity with blood supply. Several 'feed-forward' mechanisms have been described that contribute to activity-dependent vasodilation as well as vasoconstriction leading to a reduction in perfusion. Involved messengers are ionic (K+), gaseous (NO), peptides (e.g., NPY, VIP) and other messengers (PGE2, GABA, glutamate, norepinephrine) that target endothelial cells, smooth muscle cells, or pericytes. Contributions of the respective signaling pathways likely vary across brain regions or even within specific brain regions (e.g., across cortex) and are likely influenced by the brain's physiological state (resting, active, sleeping) or pathological departures from normal physiology.

The manuscript "Elevated pyramidal cell firing orchestrates arteriolar vasoconstriction through COX-2-derived prostaglandin E2 signaling" by B. Le Gac, et al. investigates mechanisms leading to activity-dependent arteriole constriction. Here, mainly working in brain slices from mice expressing channelrhodopsin 2 (ChR2) in all excitatory neurons (Emx1-Cre; Ai32 mice), the authors show that strong optogenetic stimulation of cortical pyramidal neurons is leading to constriction that is mediated through the cyclooxygenase-2 / prostaglandin E2 / EP1 and EP3 receptor pathway with contribution of NPY-releasing interneurons and astrocytes releasing 20-HETE. Specifically, using patch clamp, the authors show that 10-s optogenetic stimulation at 10 and 20 Hz leads to vasoconstriction (Figure 1), in line with a stimulation frequency-dependent increase in somatic calcium (Figure 2). The vascular effects were abolished in presence in TTX and significantly reduced in presence of glutamate receptor antagonists (Figure 3). The authors further show with RT-PCR on RNA isolated from patched cells that ~50% of analyzed cells express COX-1 or -2 and other enzymes required to produce PGE2 or PGF2a (Figure 4). Further, blockade of COX-1 and -2 (indomethacin), or COX-2 (NS-398) abolishes constriction. In animals with chronic cranial window that were anesthetized with ketamine and medetomidine, 10-s long optogenetic stimulation at 10 Hz leads to considerable constriction, which is reduced in presence of indomethacin. Blockade of EP1 and EP3 receptors leads to significant reduction of the constriction in slices (Figure 5). Finally, the authors show that blockade of 20-HETE synthesis caused moderate and NPY Y1 receptor blockade a complete reduction of constriction.

The mechanistic analysis of neurovascular coupling mechanisms as exemplified here will guide further in-vivo studies and has important implications for human neuroimaging in health and disease. Most of the data in this manuscript uses brain slices as experimental model which contrasts with neurovascular imaging studies performed in awake (headfixed) animals. However, the slice preparation allows for patch clamp as well as easy drug application and removal. Further, the authors discuss their results in view of differences between brain slices and in vivo observations experiments, including the absence of vascular tone as well as blood perfusion required for metabolite (e.g., PGE2) removal, and the presence of network effects in the intact brain. The manuscript and figures present the data clearly; regarding the presented mechanism, the data supports the authors conclusions. Some of the data was generated in vivo in head-fixed animals under anesthesia; in this regard, the authors should revise introduction and discussion to include the important distinction between studies performed in slices, or in acute or chronic in-vivo preparations under anesthesia reduced network activity and reduced or blockade of neuromodulation, or in awake animals (virtually undisturbed network and neuromodulatory activity). Further, while discussed to some extent, the authors could improve their manuscript by more clearly stating if they expect the described mechanism to contribute to CBF regulation under 'resting state conditions' (i.e., in absence of any stimulus), during short or sustained (e.g., visual, tactile) stimulation, or if this mechanism is mainly relevant under pathological conditions; especially in context of the optogenetic stimulation paradigm being used (10-s long stimulation of many pyramidal neurons at moderate-high frequencies) and the fact that constriction leading to undersupply in response to strongly increased neuronal activity seems counterintuitive?

The authors have addressed all comments, and I appreciate their insightful discussion and revision of the manuscript.

---

## [Referee Report · Reviewer #2 (Public review)]

Summary:

The present study by Le Gac et al. investigates the vasoconstriction of cerebral arteries during neurovascular coupling. It proposes that pyramidal neurons firing at high frequency lead to prostaglandin E2 (PGE2) release and activation of arteriolar EP1 and EP3 receptors, causing smooth muscle cell contraction. The authors further claim that interneurons and astrocytes also contribute to the vasoconstriction via neuropeptide Y (NPY) and 20-hydroxyeicosatetraenoic acid (20-HETE) release, respectively. The study mainly uses brain slices and pharmacological tools in combination with Emx1-Cre;Ai32 transgenic mice expressing the H134R variant of channelrhodopsin-2 (ChR2) in the cortical glutamatergic neurons for precise photoactivation. Stimulation with 470 nm light using 10-second trains of 5-ms pulses at frequencies from 1-20 Hz revealed small constrictions at 10 Hz and robust constrictions at 20 Hz, which were abolished by TTX and partially inhibited by a cocktail of glutamate receptor antagonists. Inhibition of cyclooxygenase-1 (COX-1) or -2 (COX-2) by indomethacin blocked the constriction both ex vivo (slices) and in vivo (pial artery), and inhibition of EP1 and EP3 showed the same effect ex vivo. Single-cell RT-PCR from patched neurons confirmed the presence of the PGE2 synthesis pathway. While the data are convincing, the overall experimental setting presents some limitations. How is the activation protocol comparable to physiological firing frequency? The delay (minutes) between the stimulation and the constriction appears contradictory to the proposed pathway, which would be expected to occur rapidly. The experiments are conducted in the absence of vascular "tone," which further questions the significance of the findings. Some of the targets investigated are expressed by multiple cell types, which makes the interpretation difficult; for example, cyclooxygenases are also expressed by endothelial cells. Finally, how is the complete inhibition of the constriction by the NPY Y1 receptor antagonist BIBP3226 consistent with a direct effect of PGE2 and 20-HETE in arterioles? Overall, the manuscript is well-written with clear data, but the interpretation and physiological relevance have some limitations. However, vasoconstriction is a rather understudied phenomenon in neurovascular coupling, and the present findings may be of significance in the context of pathological brain hypoperfusion.

---

## [Author Response]

The following is the authors’ response to the original reviews.

**Public Reviews:**

**Reviewer #1 (Public review):**
Neuronal activity spatiotemporal fine-tuning of cerebral blood flow balances metabolic demands of changing neuronal activity with blood supply. Several 'feed-forward' mechanisms have been described that contribute to activity-dependent vasodilation as well as vasoconstriction leading to a reduction in perfusion. Involved messengers are ionic (K+), gaseous (NO), peptides (e.g., NPY, VIP), and other messengers (PGE2, GABA, glutamate, norepinephrine) that target endothelial cells, smooth muscle cells, or pericytes. Contributions of the respective signaling pathways likely vary across brain regions or even within specific brain regions (e.g., across the cortex) and are likely influenced by the brain's physiological state (resting, active, sleeping) or pathological departures from normal physiology.The manuscript "Elevated pyramidal cell firing orchestrates arteriolar vasoconstriction through COX-2derived prostaglandin E2 signaling" by B. Le Gac, et al. investigates mechanisms leading to activitydependent arteriole constriction. Here, mainly working in brain slices from mice expressing channelrhodopsin 2 (ChR2) in all excitatory neurons (Emx1-Cre; Ai32 mice), the authors show that strong optogenetic stimulation of cortical pyramidal neurons leads to constriction that is mediated through the cyclooxygenase-2 / prostaglandin E2 / EP1 and EP3 receptor pathway with contribution of NPY-releasing interneurons and astrocytes releasing 20-HETE. Specifically, using a patch clamp, the authors show that 10-s optogenetic stimulation at 10 and 20 Hz leads to vasoconstriction (Figure 1), in line with a stimulation frequency-dependent increase in somatic calcium (Figure 2). The vascular effects were abolished in the presence of TTX and significantly reduced in the presence of glutamate receptor antagonists (Figure 3). The authors further show with RT-PCR on RNA isolated from patched cells that ~50% of analyzed cells express COX-1 or -2 and other enzymes required to produce PGE2 or PGF2a (Figure 4). Further, blockade of COX-1 and -2 (indomethacin), or COX-2 (NS-398) abolishes constriction. In animals with chronic cranial windows that were anesthetized with ketamine and medetomidine, 10-s long optogenetic stimulation at 10 Hz leads to considerable constriction, which is reduced in the presence of indomethacin. Blockade of EP1 and EP3 receptors leads to a significant reduction of the constriction in slices (Figure 5). Finally, the authors show that blockade of 20-HETE synthesis caused moderate and NPY Y1 receptor blockade a complete reduction of constriction.The mechanistic analysis of neurovascular coupling mechanisms as exemplified here will guide further in-vivo studies and has important implications for human neuroimaging in health and disease. Most of the data in this manuscript uses brain slices as an experimental model which contrasts with neurovascular imaging studies performed in awake (headfixed) animals. However, the slice preparation allows for patch clamp as well as easy drug application and removal. Further, the authors discuss their results in view of differences between brain slices and in vivo observations experiments, including the absence of vascular tone as well as blood perfusion required for metabolite (e.g., PGE2) removal, and the presence of network effects in the intact brain. The manuscript and figures present the data clearly; regarding the presented mechanism, the data supports the authors' conclusions.

We thank the reviewer for his/her supportive comments as well as for pointing out pros and cons of the brain slice preparation.

Some of the data was generated in vivo in head-fixed animals under anesthesia; in this regard, the authors should revise the introduction and discussion to include the important distinction between studies performed in slices, or in acute or chronic in-vivo preparations under anesthesia reduced network activity and reduced or blockade of neuromodulation, or in awake animals (virtually undisturbed network and neuromodulatory activity).

We have now added a paragraph in the introduction (lines 52-64) to highlight the distinction between ex vivo and in vivo models. We now also discuss that anesthetized animals exhibit slower NVC (Line 308-309).

Further, while discussed to some extent, the authors could improve their manuscript by more clearly stating if they expect the described mechanism to contribute to CBF regulation under 'resting state conditions' (i.e., in the absence of any stimulus), during short or sustained (e.g., visual, tactile) stimulation, or if this mechanism is mainly relevant under pathological conditions; especially in the context of the optogenetic stimulation paradigm being used (10-s long stimulation of many pyramidal neurons at moderate-high frequencies) and the fact that constriction leading to undersupply in response to strongly increased neuronal activity seems counterintuitive?

We now discuss more extensively the physiological relevance (lines 422-434 and 436-439) and the conditions where the described mechanisms of neurogenic vasoconstriction may occur.

We agree with the reviewer that vasoconstriction in response to a large increase in neuronal activity is counterintuitive as it leads to undersupply despite an increased energy demand. We now discuss its potential physio/pathological role in attenuating neuronal activity by reducing energy supply (lines 453-464).

**Reviewer #2 (Public review):**
Summary:The present study by Le Gac et al. investigates the vasoconstriction of cerebral arteries during neurovascular coupling. It proposes that pyramidal neurons firing at high frequency lead to prostaglandin E2 (PGE2) release and activation of arteriolar EP1 and EP3 receptors, causing smooth muscle cell contraction. The authors further claim that interneurons and astrocytes also contribute to vasoconstriction via neuropeptide Y (NPY) and 20-hydroxyeicosatetraenoic acid (20-HETE) release, respectively. The study mainly uses brain slices and pharmacological tools in combination with Emx1Cre; Ai32 transgenic mice expressing the H134R variant of channelrhodopsin-2 (ChR2) in the cortical glutamatergic neurons for precise photoactivation. Stimulation with 470 nm light using 10-second trains of 5-ms pulses at frequencies from 1-20 Hz revealed small constrictions at 10 Hz and robust constrictions at 20 Hz, which were abolished by TTX and partially inhibited by a cocktail of glutamate receptor antagonists. Inhibition of cyclooxygenase-1 (COX-1) or -2 (COX-2) by indomethacin blocked the constriction both ex vivo (slices) and in vivo (pial artery), and inhibition of EP1 and EP3 showed the same effect ex vivo. Single-cell RT-PCR from patched neurons confirmed the presence of the PGE2 synthesis pathway.While the data are convincing, the overall experimental setting presents some limitations. How is the activation protocol comparable to physiological firing frequency?

As also suggested by Reviewer #1 we have now discussed more extensively the physiological relevance of our observations (lines 422-434 and 436-439).

The delay (minutes) between the stimulation and the constriction appears contradictory to the proposed pathway, which would be expected to occur rapidly. The experiments are conducted in the absence of vascular "tone," which further questions the significance of the findings.

The slow kinetics observed ex vivo are probably due to the low recording temperature and the absence of pharmacologically induced vascular tone, as already discussed (lines 312-317). Furthermore, as recommended by reviewer #1, we have presented the advantages and limitations of ex vivo and in vivo approaches (lines 52-64).

Some of the targets investigated are expressed by multiple cell types, which makes the interpretation difficult; for example, cyclooxygenases are also expressed by endothelial cells.

Under normal conditions, endothelial cells only express COX-1 and barely COX-2, whose expression is essentially observed in pyramidal cells (see Tasic et al. 2016, Zeisel et al. 2015, Lacroix et al., 2015). As pointed out by Reviewer # 1, our ex vivo pharmacological data clearly indicate that vasoconstriction is mostly due to COX-2 activity, and to a much lesser extent to COX-1. Since it is well established that the previously described vascular effects of pyramidal cells are essentially mediated by COX-2 activity (Iadecola et al., 2000; Lecrux et al., 2011; Lacroix et al., 2015), we are quite confident that vasoconstriction described here is mainly due COX-2 activity of pyramidal cells.

Finally, how is the complete inhibition of the constriction by the NPY Y1 receptor antagonist BIBP3226 consistent with a direct effect of PGE2 and 20-HETE in arterioles?

We agree with both reviewers that the complete blockade of the constriction by the NPY Y1 receptor antagonist BIBP3226 needs to be more carefully discussed. We have now included in the discussion the possible involvement of Y1 receptors in pyramidal cells, which could promote glutamate release and possibly COX-2, thereby contributing to PGE2 and 20-HETE signaling (lines 402-409).

Overall, the manuscript is well-written with clear data, but the interpretation and physiological relevance have some limitations. However, vasoconstriction is a rather understudied phenomenon in neurovascular coupling, and the present findings may be of significance in the context of pathological brain hypoperfusion.

We thank the reviewer for his/her comment and suggestions, which have helped us to improve our manuscript.

**Recommendations for the authors:**

**Reviewer #1 (Recommendations for the authors):**
Methods:It is not clear if brain slices (or animals) underwent one, two, or several optogenetic stimulations - especially for experiments where 'control' is compared to 'treated' - does this data come from the same vessels (before and after treatment) or from two independent groups of vessels? If repeated stimulations are performed, do these repeated stimulations cause the same vascular response?

As indicated in the Materials and Methods section, line 543: “Only one arteriole was monitored per slice” implies that the comparisons between the ‘control’ and ‘treated’ groups were made from independent groups of vessels. To clarify this point, we have added “receiving a single optogenetic or pharmacological stimulation” to this sentence lines 543-544.

For in vivo experiments, animals underwent 10-20 optogenetic stimulations with a 5-minute interstimulus interval during an experiment lasting 2 hours for maximum. Trials from the same vessel were averaged (with a 0.1 s interpolation) for analysis, and the mean per vessels is presented in the graphics.

Figure 2:Can the authors speculate about the cause for the slow increase in indicator fluorescence from minute 1.5 onward, which seems dependent on stimulation frequency? Is this increase also present when slices from a ChR2-negative animal undergo the same stimulation paradigm?

Rhod2 was delivered by the patch pipette as indicated in the Materials and Methods section (line 514). Although a period of “at least 15 min after passing in whole-cell configuration to allow for somatic diffusion of the dye” (line 551-552) was observed, this single-wavelength Ca2+ indicator likely continued to diffuse into the cells during the optical recording thereby, inducing a slight increase in delta F/F0, which is consistent with the positive slopes of the mean fluorescence changes observed during the 30-s control baseline (Fig. 2b).

Figure 4: Why did the authors include panel (a) here? Also, do the authors observe that cells with different COX-1 or -2 expression profiles show different (electrical, morphological) properties?

The purpose of panel (a) in Fig. 4 was to ensure the regular spiking electrophysiological phenotype of the pyramidal neurons whose cytoplasm was harvested for subsequent RT-PCR analysis. Despite our efforts, we found no difference in the 32 electrophysiological features between COX-1 or COX-2 positive and negative cells. This is now clearly stated in the result section (lines 210-212) and a supplementary table of electrophysiological features is now provided. Because it is difficult to determine the morphology of neurons analyzed by single-cell RT-PCR (Devienne et al. 2018), these cells were not processed for biocytin labeling.

Figure 5: (1) Maybe the authors could highlight panels b-f as in vivo experiments to emphasize that these are in-vivo observations while the other experiments (especially panels g, h) are made in slices?

We thank the reviewer for this suggestion. A black frame is now depicted in Figure 5 to emphasize in vivo experiments.

(2) What is the power of the optogenetic stimulus in this experiment?

The power of the optogenetic stimulus was 38 mW/mm^2^ in ex vivo experiments (see Line 527). For in vivo experiments, 1 mW pulses of 5 ms were used, the intensity being measured at the fiber end. We now provide the information for in vivo experiments in the Methods lines 639-640.

(3) Experiments were performed with Fluorescein-Dextran at 920-nm excitation which would overlap with EYFP fluorescence from the ChR2-EYFP transgene. Did the authors encounter any issues with crosstalk between the two labels?

Crosstalk between EYFP and fluorescein fluorescence was indeed an issue. This is why arterioles were monitored at the pial level to avoid fluorescence contamination from the cortical parenchyma. Because of the perivascular space around pial arterioles, it was possible to measure vessel diameter without pollution for the parenchyma (see Author response image 1 below). To clarify this point we added the statement “which are not compromised by the fluorescence from the ChR2-EYFP transgene in the parenchyma (Madisen et al. 2012),” Line 628-629. Note that line-scan acquisitions without photoactivation stimulation did not trigger any progressive change in the vessel size or resting fluorescence.

**Author response image 1. sa3fig1:** Example of a pial arteriole filled with fluorescein dextran (cyan) in an Emx1-EYFP mouse (parenchyma labeled with YFP, in cyan). The red line represents a line-scan to record the change in diameter. Due to the perivascular space surrounding the arterioles, the vessel walls are clearly identified and separated from the fluorescent parenchyma.

(4) Could the authors potentially extend the time course in panel e to show the recovery of the preparation to the baseline?

Because arterioles were only monitored for a 40-s period during a session of optogenetic stimulation/imaging we cannot extend panel e. Nonetheless, a 5 minutes interstimulus interval was observed to allow the full recovery of the preparation to the baseline. This now clarified line 640. Of note, the arteriole shown in panel d before indomethacin treatment fully recovered to baseline after this treatment.

Also, did the authors observe any 'abnormal' behavior of the vasculature after stimulation, such as large-amplitude oscillations? (5)

We did not specifically investigate resting state oscillations, such as vasomotion, but the 10-s long baseline recording for each measurement indicates no long lasting, abnormal and de novo behavior with a frequency higher than 0.1-0.2 Hz.

Can the authors show in vivo data from control experiments in EYFP-expressing or WT mice that underwent the same stimulation paradigm (Supplementary Figure 1 shows data from brain slices)?

The reviewer is correct to point out this important control, as optogenetic stimulation can induce a vascular response without channel rhodopsin activation at high power (see our study on the topic, Rungta et al, Nat Com 2017). We therefore tested this potential artefact in a WT mouse using our setup, with different intensities and durations of optogenetic stimulation.

Author response image 2A shows that stimulations of 10 seconds, 10 Hz, 1 mW, 5 ms pulses, i.e. the conditions we used for the experiments in Emx1 mice, did not induce dilation or constriction. Stimulation for 5 seconds with the same number of pulses, but with a higher power (4 mW), longer duration (20 ms pulses) and at a higher frequency elicited a small dilation in 1 of 2 pial arterioles (Author response image 2B). For this reason, we used only shorter (5ms) and less intense (1 mW) optogenetic stimulation to ensure that the observed dilation was solely due to Emx1 activation and not to light-induced artefactual dilation.

**Author response image 2. sa3fig2:** Optogenetic stimulation in a wild-type mouse. A. No diameter changes upon stimulations of 10 seconds, 10 Hz, 1 mW, 5 ms pulses, i.e. the conditions we used for the experiments in Emx1 mice. B. Stimulation of higher power (4 mW), longer duration (20 ms pulses) and at a higher frequency elicited a small dilation in 1 (grey traces) of 2 pial arterioles.

Figures 6 and 7: It is surprising that blockade of NPY Y1 receptors leads to a complete loss of the constriction response. As shown in Figure 7, the authors suggest that pyramidal neuron-released PGE2 (and glutamate) initiate several cascades acting on smooth muscle directly (PGE2-EP1/EP3), through astrocytes (Glu/COX-1/PGE2 or 20-HETE), or through NPY interneurons (Glu/NPY/Y1 or PGE2/NPY/Y1). This would imply that COX-1/2 and NPY/Y1 pathways act in series (as discussed by the authors). Besides the potential effects on NPY release mentioned in the discussion, could the authors comment if both (NPY and PGE2) pathways need to be co-activated in smooth muscle cells to cause constriction?

We thank the reviewer for raising this surprising complete loss of vasoconstriction by Y1 antagonism, despite the contribution of other vasoconstrictive pathways. We now discuss (lines 402-409) the possibility that activation of the neuronal Y1 receptors in pyramidal cells may also have contributed to the vasoconstriction by promoting glutamate and possibly PGE2 release. The combined activation of vascular and neuronal Y1 receptors may explain the complete blockage of optogenetically induced vasoconstriction by BIBP3226.

**Reviewer #2 (Recommendations for the authors):**
The complete block of the constriction by BIBP3226 needs to be carefully considered.

We thank the reviewer for stressing this point also raised by Reviewer #1. As mentioned above we now discuss (lines 402-409) the possibility that activation of the neuronal Y1 receptors in pyramidal cells may also have contributed to the vasoconstriction by promoting glutamate and possibly PGE2 release. The combined activation of vascular and neuronal Y1 receptors may explain the complete blockage of optogenetically induced vasoconstriction by BIBP3226.